

# Reconstructing atmospheric H₂ over the past century from bi-polar firn air records

John D. Patterson[1], Murat Aydin[1], Andrew M. Crotwell[2,3], Gabrielle Pétron[2,3], Jeffery P. Severinghaus[4], Paul B. Krummel[5], Ray L. Langenfelds[5], Vasilii V. Petrenko[6], and Eric S. Saltzman[1]

[1]Department of Earth System Science and Chemistry, University of California, Irvine, Irvine, CA 92697, USA
[2]Cooperative Institute for Research in Environmental Sciences, University of Colorado, Boulder, Boulder, CO 80309, USA
[3]Global Monitoring Laboratory, National Oceanic and Atmospheric Administration, Boulder, CO 80305, USA
[4]Scripps Institution of Oceanography, University of California, San Diego, La Jolla, CA 92093, USA
[5]Climate Science Centre, Commonwealth Scientific and Industrial Research Organisation, Environment, Aspendale, Victoria 3195, Australia
[6]Department of Earth and Environmental Sciences, University of Rochester, Rochester, NY 14627, USA

*Correspondence to*: John D. Patterson (jdpatter@uci.edu)

## Abstract

Historical hemispheric atmospheric H₂ levels since 1930 were reconstructed using the UCI_2 firn air model and firn air measurements from three sites in Greenland: (NEEM, Summit, and Tunu) and two sites in Antarctica (South Pole and Megadunes). A joint reconstruction based on the two Antarctic sites yields H₂ levels monotonically increasing from about 350 ppb in 1900 to 550 ppb in the late 1990's, levelling off thereafter. These results are similar to individual reconstructions published previously (Patterson et al., 2020; 2021). Reconstruction of the Greenland data is complicated by a systematic bias between Tunu and the other sites. The Tunu reconstruction shows substantially lower historical H₂ levels than the other two sites, a difference we attribute to possible bias in the calibration of the Tunu measurements. All three reconstructions show a late 20th century maximum in H₂ levels over Greenland. A joint reconstruction of the Greenland data shows H₂ levels rising 40% from 1930-1990, reaching a maximum of 550 ppb. After 1990, reconstructed atmospheric H₂ decrease by 6% over the next 20 years. The reconstruction deviates by at most 4% from the few available surface air measurements of atmospheric H₂ levels over Greenland from 1998-2004. However, the longer instrumental records from sampling sites outside of Greenland show a more rapid decrease and stabilization after 1990 compared to the reconstruction. We explore the possibility that this difference is an artefact caused by the firn air model underestimating pore close-off induced enrichment, evidenced by a mismatch between measured and modelled Ne in firn air. We developed new parameterizations which more accurately capture pore close-off induced enrichment at the Greenland sites. Incorporating those parameterizations into the UCI_2 model yields reconstructions with lower H₂ levels throughout the mid-late 20th century and more stable H₂ levels during the 1990's, in better agreement with the flask measurements.



# 1 Introduction

Molecular hydrogen ($H_2$) is the second most abundant reduced gas in Earth's atmosphere after methane. Atmospheric $H_2$ levels are linked to Earth's radiative budget, air quality, and the atmospheric oxidative capacity (eg. Ehhalt & Rohrer, 2009; Novelli et al., 1999; Paulot et al., 2021). As $H_2$ becomes a more important component of the energy sector, anthropogenic emissions of $H_2$ are expected to rise substantially (Derwent et al., 2020; Prather, 2003; Wang et al., 2013a, 2013b). Projecting the atmospheric response to increased anthropogenic emissions in a changing climate requires a comprehensive understanding of the biogeochemical cycle of $H_2$. Studying past changes in atmospheric $H_2$ can inform predictions of the effects of future perturbations by providing insight into the relationship between atmospheric $H_2$, human activities, and climate.

The globally averaged mixing ratio of $H_2$ in the modern atmosphere is roughly 530 nmol $mol^{-1}$ (ppb). Major sources of atmospheric $H_2$ include direct emissions from fossil fuel combustion and biomass burning and photochemical production from the oxidation of methane and non-methane hydrocarbons (NMHCs). Direct emission as a by-product of $N_2$ fixation is a minor source of atmospheric $H_2$. The major sink of atmospheric $H_2$ is consumption by soil microbes, accounting for about 70% of total losses and the remainder of the losses are to oxidation by the OH radical. Based on the atmospheric burden and estimates of the microbial and OH sinks, the lifetime of $H_2$ is estimated at 2 years (Novelli et al., 1999; Pieterse et al., 2011; 2013; Paulot et al., 2021).

Increasing atmospheric $H_2$ could influence the climate system in several ways. The reaction of $H_2$ with OH represents a loss for OH and therefore increases the atmospheric methane lifetime and associated radiative forcing. This reaction leads to the production of $HO_2$, which can influence tropospheric ozone levels. In the stratosphere, the reaction of $H_2$ with OH leads to the production of water vapor. Increased stratospheric water vapor cools the stratosphere and warms the troposphere. Paulot et al. (2021) estimated the effective radiative forcing of atmospheric $H_2$ at 0.13 mW $m^{-2}$ $ppb^{-1}$ due to its effects on the methane lifetime and stratospheric water vapor content.

Systematic measurements of atmospheric $H_2$ levels began in the late 1980's (Khalil & Rasmussen, 1990). The NOAA Global Monitoring Laboratory (NOAA/GML), CSIRO GASLAB (CSIRO), and the Advanced Global Atmospheric Gases Experiment (AGAGE) started to monitor atmospheric $H_2$ levels at several sites around the world in the early 1990's (Novelli et al., 1999; Novelli, 2006; Prinn et al., 2019; Langenfelds et al., 2002; Pétron et al., 2023). Integration of records produced by the different groups has been complicated by calibration issues. Earlier use of different scales was addressed by implementation of the internationally accepted MPI09 scale established by the Max Planck Institute for Biogeochemistry (MPI; Jordan & Steinberg, 2011). Measurements from CSIRO and AGAGE, have been transferred to the MPI09 scale. Additionally, NOAA/GML measurements from 2009 onward were recently revised to the MPI09 scale (Pétron et al., 2023). Broadly, the instrumental record shows northern hemispheric $H_2$ levels rising during the late 1980's to a maximum near 1990. There is no discernible trend in Northern Hemisphere $H_2$ levels during the 1990's and the first decade of the 2000's





(Figure 3). The Southern Hemisphere instrumental record shows $H_2$ levels rising until 1999, then plateauing for the next 10-15 years.

Longer-term changes in atmospheric $H_2$ levels during the 20th century have been reconstructed using polar firn air from Antarctica (South Pole, Megadunes) and Greenland (NEEM; Petrenko et al., 2013; Patterson et al., 2020; 2021). Firn is the compacted snow layer that forms the upper 40 – 120 m of ice sheets and its interconnected porosity contains an atmospheric archive that can extend up to ≈100 years back in time. For a comprehensive introduction to firn and firn air we refer the interested reader to Buizert (2013).  The Antarctic reconstructions show that $H_2$ levels increased by 60% over the 20th century, consistent with the instrumental record and with increasing anthropogenic emissions and photochemical production from methane (Patterson et al., 2020). The Greenland reconstruction shows an increase in atmospheric $H_2$ levels during the 1960's to a peak in the late 1980's or early 1990's, followed by a small decline from 1990-2009 (Petrenko et al., 2013). This recent decline is inconsistent with northern hemisphere modern flask measurements that show roughly constant $H_2$ levels after the mid-1990's (Novelli, 2006; Prinn et al., 2019). Petrenko et al. (2013) noted that the firn air model used for that reconstruction did not include pore close-off fractionation, a process that enriches $H_2$ in the lock-in zone where vertical diffusion of trace gases effectively goes to zero. Patterson et al. (2020) suggested that the inferred peak could be an artifact of ignoring this process.

In this work, we reassess historical Northern Hemisphere atmospheric $H_2$ levels over the 20th century using firn air from NEEM and two additional Greenland sites with previously unpublished $H_2$ measurements (Summit and Tunu). Reliable bipolar records are useful for constraining changes to $H_2$ cycling because the two hemispheres differ in their sensitivity to changes in the soil sink and anthropogenic emissions. Understanding past variability in the soil sink is particularly important because the response of the soil sink to future changes in climate is the largest source of uncertainty in projecting the radiative consequences of increasing anthropogenic $H_2$ emissions (Warwick et al., 2022).

## 2. $H_2$ measurements in Greenland firn air

### 2.1 Firn air sampling

The sampling methods used in the Summit, Tunu, and NEEM campaigns were similar to techniques from previous published firn air studies (e.g. Battle et al., 1996; Severinghaus et al., 2001; Severinghaus & Battle, 2006). Briefly, a borehole was drilled into the firn, pausing the drilling at each desired sampling depth. The borehole was sealed above each sampling depth with an inflatable rubber packer to prevent contamination from the modern atmosphere. Synflex 1300 (formerly known as "Decabon") tubing for both waste air and sample air extends from below the packer to the surface. The waste air intake was positioned directly below the rubber packer. The waste air intake was separated from the sample air intake by a stainless-steel plate (baffle) with a diameter slightly less than the borehole. Air was pumped from the waste air intake 2-5x faster than from the sample air intake to ensure that no air that had been in contact with the rubber packer was



sampled. Sampled air was dried using $Mg(ClO_4)_2$ and stored in 2.5 L glass flasks. More detailed information regarding the
sampling at each site may be found in the references listed in Table 1.


**Table 1-** Site characteristics and data references for the three Greenland firn air sampling campaigns.

| Site | Lat/lon | Date | T (ºC)[1,4] | Accum.[2,4] | Ref. | Firn air model parameters[3] |
|------|---------|------|-------------|-------------|------|------------------------------|
| NEEM | 77ºN, 51ºW | 7/2008 | -29 | 21.6 | Petrenko et al., 2013 | Buizert et al., 2012 |
| Summit | 73ºN, 38ºW | 5/2013 | -31 | 23.4 | Hmiel et al., 2020 | C. Buizert, personal communication |
| Tunu | 78º N, 34º W | 5/1996 | -29 | 10.0 | Butler et al., 1999 | This study |

[1]Mean annual surface temperature
[2]Average modern accumulation (cm ice equivalent per year)
[3]Modeling parameters include density profile, diffusivity profile, partitioning between open and closed porosity, and the
lock-in depth (Section 3.1).
[4]Temperature and accumulation for NEEM and Tunu are from Buizert et al. (2012) and Butler et al. (1999) respectively.
Temperature and accumulation for Summit were supplied by Christo Buizert (Personal Communication).

**2.2 Firn air measurements and calibration**

Firn air samples from Tunu were analyzed for $H_2$ by the NOAA/GML Carbon Cycle Group using a reducing gas
analyzer (RGA) with HgO detection.  At the time of analysis, the measurements were calibrated using a single 530 ppb $H_2$
standard assuming a linear detector response. We subsequently applied an empirical correction to account for the non-linear
response of the HgO-RGA. The non-linearity bias is estimated to be -3 ppb at 453 ppb and -12 ppb at 312 ppb based on
laboratory measurements made at NOAA/GML in 2009. The biases were linearly interpolated onto the firn air measurements
(i.e.  a measurement of 300 ppb was corrected to 312 ppb). Jordan & Steinberg (2011) found a substantially larger non-
linearity bias for a similar RGA detector. The sensitivity of our results to uncertainty in the non-linearity correction is
minimal as demonstrated in the Supplement (Figures S3 and S4).  The Tunu data were originally calibrated on the NOAA96
calibration scale, that is known to have drifted due to $H_2$ increasing in the GML gravimetric standards stored in aluminum
cylinders. We adjusted the Tunu firn air measurements to the MPI09 scale based on a matched flask intercomparison
between NOAA/GML and CSIRO conducted using atmospheric samples from Cape Grim Observatory, Tasmania during
June and July 1996.  The intercomparison gives an average offset (CSIRO-NOAA/GML) of 4.7 ppb. A constant correction
of 4.7 ppb was added to the Tunu measurements to account for the offset. The Summit firn air measurements were made by
NOAA/GML using a He pulsed discharge detector with linear response (HePDD; Novelli et al., 2009). The Summit





measurements have been formally revised to the MPI09 calibration scale. For both the Tunu and Summit data, the corrections described above are below 5% and do not significantly influence the main conclusions of this research.

Firn air samples from NEEM were analyzed for $H_2$ at CSIRO using gas chromatography coupled to an HGO-RGA. These measurements were calibrated using standards between 340 and 1000 ppb. The CSIRO measurements have been revised to the MPI09 scale. CSIRO estimates the analytical uncertainty to be ±0.2%. However, there are other sources of

uncertainty in the true $H_2$ content in the firn air. For example, tests conducted during the NEEM campaign demonstrated a small procedural blank (4-6 ppb). No such tests were conducted at the other sites. To account for the possibility of such blanks, we assign an overall uncertainty of ±2% to the data from all sites.

The firn air $H_2$ measurements from all three sites were corrected for gravitational fractionation (equation 2; Section 3.1), and duplicate measurements at the same depth averaged to generate the finalized depth profiles (Figure 1b-d; 2b-d).

Lock-in depths for the sites are given in Table 2. The $H_2$ depth profiles at the three Greenland sites display strong gradients above lock-in that reflect the fast gas phase diffusivity of $H_2$ and intense seasonality of atmospheric $H_2$ in the northern hemisphere. The $H_2$ gradients observed in the upper firn at NEEM are different than the gradients at Summit and Tunu because NEEM was sampled in July while the other two sites were sampled in May. The depth profiles from NEEM and Summit both display sharp maxima of 530 and 560 ppb respectively in the lock-in zone. In contrast, Tunu $H_2$ levels do not

display a peak in the lock-in zone. Instead, the Tunu $H_2$ measurements are approximately constant for the 10 m just above lock-in and then decrease significantly to a minimum of 420 ppb at the bottom of lock-in. The NEEM measurements also show a sharp decrease to 490 ppb at the bottom of lock-in. At Summit, the decrease towards the bottom of lock-in is less dramatic than at the other sites because only one sample depth is deeper than the observed maximum. The qualitative differences in the depth profiles reflect both the different sampling times ranging from 1996-2013 and the different site

physical characteristics. For example, the firn air at the bottom of the lock-in zone at Summit is younger than the two other sites because: 1) Summit was sampled later than the other sites and 2) Summit has a much higher accumulation rate than Tunu and a thinner lock-in zone than NEEM (Table 1).

**Table 2-** Summary of firn air $H_2$ measurements from NEEM, Summit, and Tunu.

| Site | Samples | Unique Depths | Lab | Detector | Lock-in depth (m) |
|------|---------|---------------|-----|----------|-------------------|
| NEEM | 23 | 18 | CSIRO | HgO-RGA | 63 |
| Summit | 37 | 19 | NOAA | HePDD | 68.5 |
| Tunu | 48 | 22 | NOAA | HgO-RGA | 58 |



## 3. Firn air modelling and inversions

### 3.1 Firn air model

The UCI_2 firn air model is a 1-dimensional finite-difference advective-diffusive model that is used to simulate the evolution of $H_2$ levels in firn air. The model domain is divided into an upper "diffusive zone" and lower "lock-in zone." In the diffusive zone, vertical gas transport occurs via wind-driven convective mixing in the shallowest ~5 m and via molecular diffusion throughout. Diffusive mixing decreases with depth due to the increasing tortuosity of the firn. In the lock-in zone, vertical molecular diffusion ceases due to the presence of impermeable ice layers. Gas transport in the lock-in zone occurs primarily due to advection with a small non-fractionating mixing term. The model uses a forward Euler integration scheme and a time step of 324 s. The model is largely based on Severinghaus et al. (2010), with two important differences: 1) thermal diffusion is neglected, as it is unimportant for $H_2$, and 2) our model parameterizes pore close-off fractionation differently than the Severinghaus model (see below). The model tracks the air content and composition in both open pores and closed bubbles as a function of time and depth. The model code is written and executed in MATLAB R2022a (Mathworks Inc.).

The site-specific bulk density profile ($\rho_{firn}$: kg m$^{-3}$) is calculated from an empirical fit to density measurements of the firn core. Total porosity ($s_{total}$; dimensionless) is estimated from the density profile:

$$s_{total} = 1 - \frac{\rho_{firn}}{\rho_{ice}} \tag{1}$$

Where $\rho_{ice}$ (kg m$^{-3}$) is the temperature dependent density of ice from Bader (1964). Accumulation rate, temperature, and the depth of the lock-in zone onset are site-specific parameters, and we implement previously published parameterizations for the partitioning between open and closed porosity at each site (Tables 1 and 2).

Model grid spacing is 0.5 m in the diffusive zone. Gas transport is dominated by convective mixing in the upper part of the diffusive zone, and by molecular diffusion in the lower part of the diffusive zone. To simulate non-fractionating convective mixing, a depth-dependent eddy diffusivity term is added to the classical one-dimensional firn air transport equation (Schwander et al., 1993; Severinghaus et al., 2010; Trudinger et al., 1997). Additionally, we neglect the typical gravitational term, and instead empirically correct the firn air measurements using the $\delta^{15}N$ of $N_2$ depth profile. The correction is calculated from the $\delta^{15}N$ data for each borehole according to equation 2:

$$Corr_{grav} = \frac{\delta^{15}N}{1000} * \frac{\exp\left(g*z*\frac{\Delta m_g}{RT}\right)-1}{\exp\left(g*z*\frac{\Delta m_{15N}}{RT}\right)-1} \tag{2}$$

Where $Corr_{grav}$ is the depth dependent fractional correction for the gas of interest, $\delta^{15}N$ is the measured isotopic composition of $N_2$ (‰) each depth, $g$ is the gravitational acceleration constant (9.8 m s$^{-2}$), $z$ is depth (m), $R$ is the ideal gas constant (8.314





J mol$^{-1}$ K$^{-1}$), $T$ is the annual average temperature at the site, $\Delta m_g$ is the difference in molar mass (in kg) between the gas of interest and air, and $\Delta m_{15N}$ is the difference in molar mass between $^{28}N_2$ and $^{29}N_2$ (.001 kg).

Equation 3 governs the evolution of the concentration of the gas of interest in the open pores in the diffusive zone
(Severinghaus et al., 2010):

$$s_o \frac{\partial c}{\partial t} = \frac{\partial}{\partial z}\left(\frac{\partial c}{\partial z} * \left[s_o D_{mol}(z,T,P) + s_o D_{eddy}(z)\right]\right) - s_o w \frac{\partial c}{\partial z} \qquad (3)$$

Where $s_o$ is the open porosity, $C$ is concentration of the gas of interest (mol m$^{-3}$), $D_{mol}$ is the gas, depth, temperature,
and pressure dependent molecular diffusivity constant (m$^2$ s$^{-1}$), $D_{eddy}$ is the depth dependent eddy diffusivity (m$^2$ s$^{-1}$), and $w$ is the downward velocity of the firn column (m s$^{-1}$). We used $D_{mol}$ and $D_{eddy}$ profiles for NEEM and Summit developed previously using a suite of measured gases with well constrained atmospheric histories of CO$_2$, CH$_4$, CH$_3$CCl$_3$, and SF$_6$ (Buizert et al., 2012; Hmiel et al., 2020; C. Buizert, personal communication). The diffusivity profiles were validated by forcing the model with the established atmospheric histories for the previously mentioned trace gases and comparing the
model output to the measurements. Diffusivity profiles for Tunu were tuned using CO$_2$ and CH$_4$ measurements. During the tuning process, the ratio between the effective molecular diffusivities of each trace gas at each depth is forced to remain identical to the ratio of their gas phase diffusivities (i.e. at each depth $D_{mol}$ for CH$_4$ is 1.4 time that of $D_{mol}$ for CO$_2$). After tuning $D_{mol}$ is then scaled using the ratio of the gas phase diffusivity of CO$_2$ to the gas phase diffusivity of the gas of H$_2$ or other trace gas of interest. Temperature and pressure dependent gas phase diffusivities are calculated using Fuller et al.
(1966) as described by Reid et al. (1987; Ch. 11).

Model grid spacing in the lock-in zone is 1 annual layer. There is no molecular diffusivity in the lock-in zone and gas transport occurs via downward advection of annual layers once per year. Small values of eddy diffusivity are prescribed in the upper part of lock-in. This non-fractionating mixing term is included to account for vertical airflow caused by barometric pressure fluctuations. Model parameterizations for vertical mixing in the lock-in zone are validated by forcing the
model with best-estimate Greenland atmospheric histories for various trace gases with well constrained atmospheric histories (Buizert et al., 2012).

In addition to convective mixing, diffusion and advection, pore close-off fractionation is parameterized in the firn air model. As closed pores (bubbles) are advected downward in the firn, internal pressure increases, creating a partial pressure gradient which drives a net diffusive flux of highly mobile trace gases out of the bubbles, through the ice lattice,
and into the open pores. The open pores are therefore enriched in these gases. This phenomenon affects gases with kinetic diameters (KD) <3.6 Å such as Ne, and He due to their large diffusivity in ice (Severinghaus & Battle, 2006).

Given its small molecular diameter (KD= 2.89 Å) pore close-off fractionation must affect H$_2$ in a similar manner to Ne (Patterson et al., 2020; Petrenko et al., 2013, Severinghaus & Battle, 2006). Recent laboratory measurements indicate that the permeability of H$_2$ in ice is sufficiently fast to equalize the partial pressure of H$_2$ in the open porosity and closed



bubbles (Patterson & Saltzman, 2021). Therefore, in the model, it is assumed that the partial pressure of $H_2$ is in equilibrium

between the closed bubbles and open pores in each layer as described by equations 4-6:

$$P_n = (P_{bubble} x_{n(bubble)} s_c + P_{ambient} x_{n(firn)} s_o)/s_{total} \qquad (4)$$

$x_{n(bubble)} = P_n/P_{bubble} \qquad (5)$

$$x_{n(firn)} = P_n/P_{ambient} \qquad (6)$$

Where $P_n$ (Pa) is partial pressure of gas $n$ ($H_2$ or Ne; See section 5), $x_n$ is mole fraction of gas $n$, $P_{bubble}$ (Pa) is the total bubble

pressure, $P_{ambient}$ (Pa) is the ambient pressure in the open pores, and $s_c$ is closed porosity. The subscripts *bubble* and *firn*

distinguish between the closed and open porosity. Equations 4-6 are executed at every time step in each grid cell in the

model. Note that the equilibrium parameterization of pore close-off fractionation used here is different from the kinetic

parameterization used by Severinghaus & Battle (2006) which was developed assuming a much smaller permeation constant

for Ne. Pore close-off fractionation is important in terms of the exchange of $H_2$ and Ne between closed and open pores.

Permeation of these gases through the ice lattice vertically between model layers can be neglected on time scales relevant to

firn air modelling (Patterson & Saltzman 2021).

### 3.2 Inversion methods

       The firn air measurements are inverted to recover an atmospheric history of $H_2$ from each site. The inversion

technique employs depth-dependent age distributions or "Green's functions" (*G*; (Rommelaere et al., 1997). The firn air

model is initialized with no $H_2$ in the firn air column, then forced with a transient 1-year pulse of $H_2$ at the surface. The

model is integrated for 300 years, and the evolution of the pulse is tracked as a function of depth and time to produce the

Green's functions. The age distributions grow older and broader with depth (Figure S1). Given an atmospheric history

($H_{2(atm)}$), modelled levels of $H_2$ in the firn ($H_{2(firn)}$) are calculated according to equation 7:


$$H_{2(firn)} = \sum_{t=0}^{300} H_{2(atm)} * G \qquad (7)$$

Using Green's function allows for rapid iteration over many possible atmospheric histories without running the firn air

model in forward mode. The inversion problem is under-constrained. Multiple atmospheric histories can provide good

agreement with the measured depth profile, and additional constraints are required to yield a unique solution. In previous

studies, the additional constraint has been implemented as a smoothing parameter (Patterson et al., 2020, 2021; Petrenko et



al., 2013; Rommelaere et al., 1997). In this work, we rely instead on the assumption of autocorrelation and utilize the probabilistic modelling software package, Stan (mcstan.org). Stan has been used previously for firn air reconstructions by Aydin et al. (2020). As in that work, we use Stan to implement a Bayesian hierarchical model in which the atmospheric
history is treated as an auto-correlated random variable:

$$m_{atm}[t_i] \sim N(m_{atm}[t_{i-1}], \beta\, m_{atm}[t_{i-1}]) \tag{8}$$

Where $m_{atm}$ is a vector of length $i$ that contains the discrete atmospheric $H_2$ dry air mole fraction history. The atmospheric
mixing ratio at time $t_i$ is normally distributed around the atmospheric mixing ratio at time $t_{i-1}$ with a standard deviation of $\beta\, m_{atm}[t_{i-1}]$. $\beta$ is a positive scalar which may be specified or varied as a free parameter. The atmospheric history is related to the firn air measurements:

$$h_{obs} \sim N(Gm_{atm}, \sigma) \tag{9}$$


Where $h_{obs}$ is a vector of length $n$, which contains the measured $H_2$ levels at each depth, $G$ is an $n$ by $i$ matrix where each row corresponds to the Green's function for that unique depth (referred to as the "kernels" by Rommeleaere et al. (1997)), and $\sigma$ is a vector of length $n$, which contains the analytical uncertainties of the firn air measurements. Stan samples the joint posterior probability distribution for the parameters, $m_{atm}$ and $\beta$ using a fast Hamiltonian Markov Chain Monte Carlo
algorithm (Carpenter et al., 2017). Parameters are sampled from uniform prior distributions unless otherwise specified. Inversions are carried out using the MatlabStan 2.15.1.0 interface to cmdstan 2.27.0. The advantages of this method over other methods include: 1) No artificial smoothing criteria is imposed (instead the atmospheric history is assumed to be autocorrelated), and 2) Stan allows us to easily explore sensitivity to assumptions about the data by encoding those assumptions as priors.
In Greenland, $H_2$ levels in the upper firn layers are strongly influenced by seasonal variations and firn air reconstructions cannot accurately reconstruct such high frequency variability. Therefore, measurements from the upper part of the firn were excluded from the reconstructions. A cut-off depth was determined by forcing the firn air model in forward mode with an atmospheric history that consisted of constant annually averaged $H_2$ levels with a realistic seasonal cycle imposed. Depths where the $H_2$ concentration varied by more than 1% from the annual average were excluded from the
reconstruction. The cut-off depths were 62 m, 66 m, and 50 m for NEEM, Summit and Tunu respectively. Unlike Greenland, Antarctic $H_2$ reconstructions are not sensitive to seasonality primarily because seasonal variability in atmospheric $H_2$ levels over Antarctica is less than half that of seasonal variability over Greenland.



## 4. Atmospheric reconstructions

Atmospheric $H_2$ reconstructions were carried out for each site independently (Figure 1). The NEEM, Summit, and Tunu sites have different physical characteristics (mean annual temperature and accumulation rate) and were sampled at different times. The firn air age distributions at the three sites are shown in Figure S1. The model inversions yield generally good agreement between the modeled and measured depth profiles for NEEM and Tunu (Figure 1b-d). Agreement is also good at Summit, except at the bottom of lock-in where the modeled depth profile increases monotonically with depth while 285     the deepest measurement shows a slight decrease.

Firn air measurements from NEEM constrain atmospheric $H_2$ levels after 1950. The NEEM reconstruction shows atmospheric $H_2$ increasing from 1950-1989 at an average rate of 3.1 ppb $y^{-1}$ (425-545 ppb). After 1989, atmospheric levels decrease at an average rate of 1.6 ppb $y^{-1}$ reaching 515 ppb in 2008. Summit firn air is younger, yielding an atmospheric history beginning in 1975. The Summit reconstruction shows atmospheric $H_2$ increasing from 505-535 from 1975 to 1994, 290     an average rate of 1.5 ppb $y^{-1}$. After 1994 there is a slight decrease in atmospheric $H_2$ levels until 2003. After 2003, atmospheric $H_2$ decreases more rapidly at a rate of 1.5 ppb $y^{-1}$, reaching 515 ppb in 2013. Tunu firn air is considerably older, showing a rise in atmospheric $H_2$ from 370-510 ppb from 1930 to 1992, an average rate of 2.3 ppb $y^{-1}$. After 1992, $H_2$ levels decreased at a rate of 1.5 ppb $y^{-1}$ to just over 500 ppb in 1996.

All three reconstructions show $H_2$ levels rising during the mid-late 20[th] century with a maximum in atmospheric $H_2$ 295     near 1990. The maximum is somewhat later and broader in the Summit reconstruction compared to the other two sites. There is only one measurement below the observed $H_2$ maximum at Summit, and that single measurement does not provide a sufficiently strong constraint on the reconstruction to generate the sharper maximum observed in the NEEM and Tunu reconstructions. We conducted another reconstruction using the Summit data with the uncertainty on the measurements reduced by 25%. Reducing the uncertainty on the Summit measurements imposes tighter constraints on the reconstruction 300     and forces better agreement between the modeled and measured depth profiles. This reconstruction shows an earlier maximum (1992) and a significantly more rapid rise in $H_2$ levels from 1975-1992, in better agreement with the reconstructions from NEEM and Tunu (Figure S2). The reconstructed decrease during the mid to late-1990's is still somewhat slower than in other two reconstructions.

Although the trends observed in the NEEM and Tunu reconstructions are in good agreement, there is an offset of 305     roughly 35 ppb in absolute reconstructed $H_2$ levels (Figure 1a). We suspect that the offset is due to a calibration problem in the Tunu data. Those measurements were made in 1996, the same year that NOAA revised their $H_2$ working standard upward by 6.4%. Revising the Tunu measurements upward by 6.4% yields a reconstruction in near perfect agreement with the NEEM reconstruction. Furthermore, the probability model suggests that adjusting the Tunu measurements upwards by 7.1% yields the best agreement between the three sites (see below), in good agreement with the 6.4% calibration adjustment. 310     The original NOAA laboratory notes for the Tunu samples indicate that the upward revision to the working standard was



applied to the Tunu measurements but the data suggest that this revision was not actually applied. An alternative explanation for the offset between sites could be *in situ* $H_2$ production (see Section 8).

A "joint" reconstruction was carried out in which the most probable atmospheric history given the firn air data from all three sites is calculated (Figure 2). To account for the apparent offset between Tunu and the other sites, we introduced
another free parameter, $\gamma$ , which is a dimensionless scalar multiplier for the data from Tunu. $\gamma$ is drawn from a uniform prior between 0.9 and 1.1. Equation 9 is adapted for the measurements from Tunu.

$$\gamma \boldsymbol{h_{obs}} \sim N(\boldsymbol{Gm_{atm}}, \boldsymbol{\sigma}) \tag{10}$$

The posterior distribution for $\gamma$ is approximately normally distributed around 1.071 with a standard deviation of 0.012, implying an offset of about 7.1% for the Tunu measurements. Atmospheric $H_2$ levels increase from 390 ppb in 1930 to 545 ppb in 1990 at an average rate of 2.6 ppb $y^{-1}$. After 1990, atmospheric $H_2$ starts to decrease at an average rate of 1.4 ppb $y^{-1}$, reaching 515 ppb in 2013. There is good agreement between the modeled and measured depth profiles for the joint reconstruction.


**Figure 1: Atmospheric histories reconstructed independently from firn air profiles at three Greenland sites. a) purple line and shading- result from NEEM and associated ±1σ uncertainty; orange line and shading- result from Summit and associated ±1σ uncertainty; dark red line and shading- result from Tunu and associated ±1σ uncertainty black x's– annual mean synthetic Summit $H_2$ history (Section 5; Pétron et al., 2023; Langenfelds et al., 2002); b) black**
**markers- measured $H_2$ depth profile at NEEM; squares with error bars are measurements used in the reconstruction, and circles are measurements excluded from the reconstruction because of seasonality ; purple line and shading-modeled depth profile using the atmospheric history plotted in purple in a) with the propagated ±1σ uncertainty; the dashed black line indicates the top of lock-in zone c) and d) as in b) for Summit and Tunu respectively**





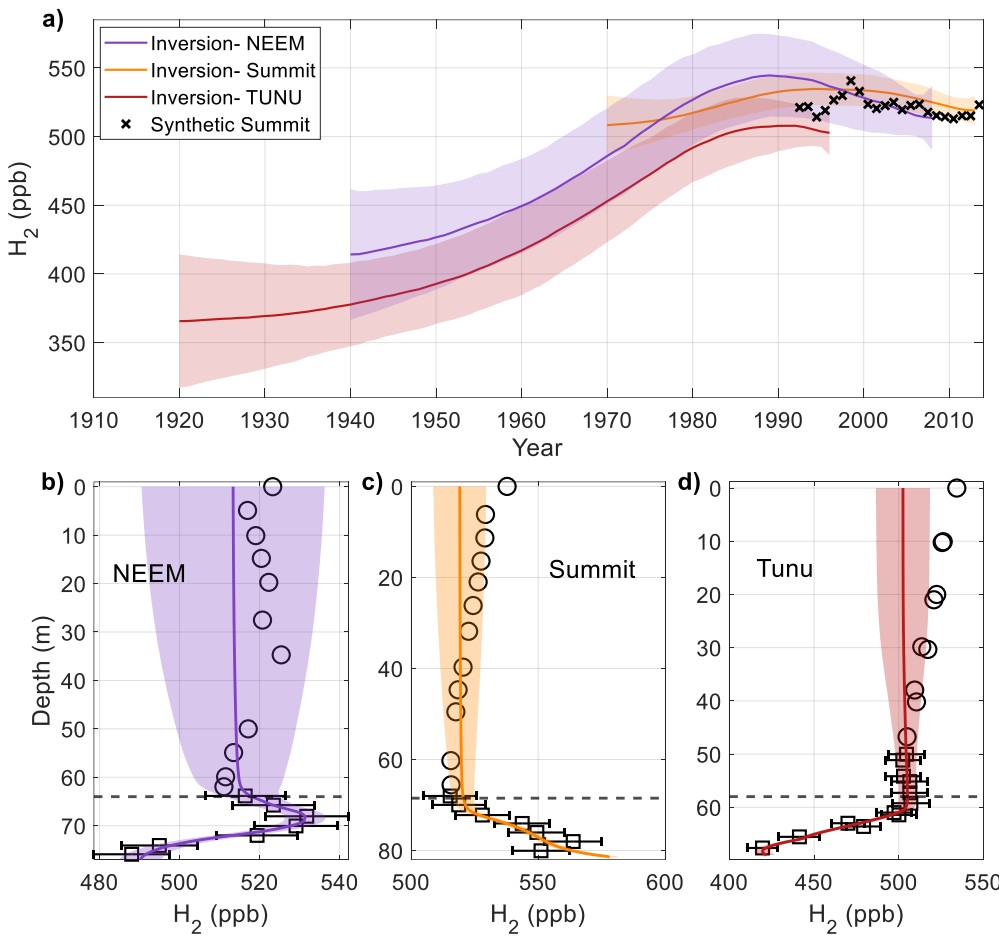


**Figure 2: Joint atmospheric H₂ reconstruction using firn air from three Greenland sites (NEEM, Summit, and Tunu). The reconstruction was conducted using equation 9 for NEEM and Summit, and equation 10 for Tunu (see text). All lines and markers are the same as in Figure 1. The Tunu measurements are adjusted upwards by 7.1% (see text).**






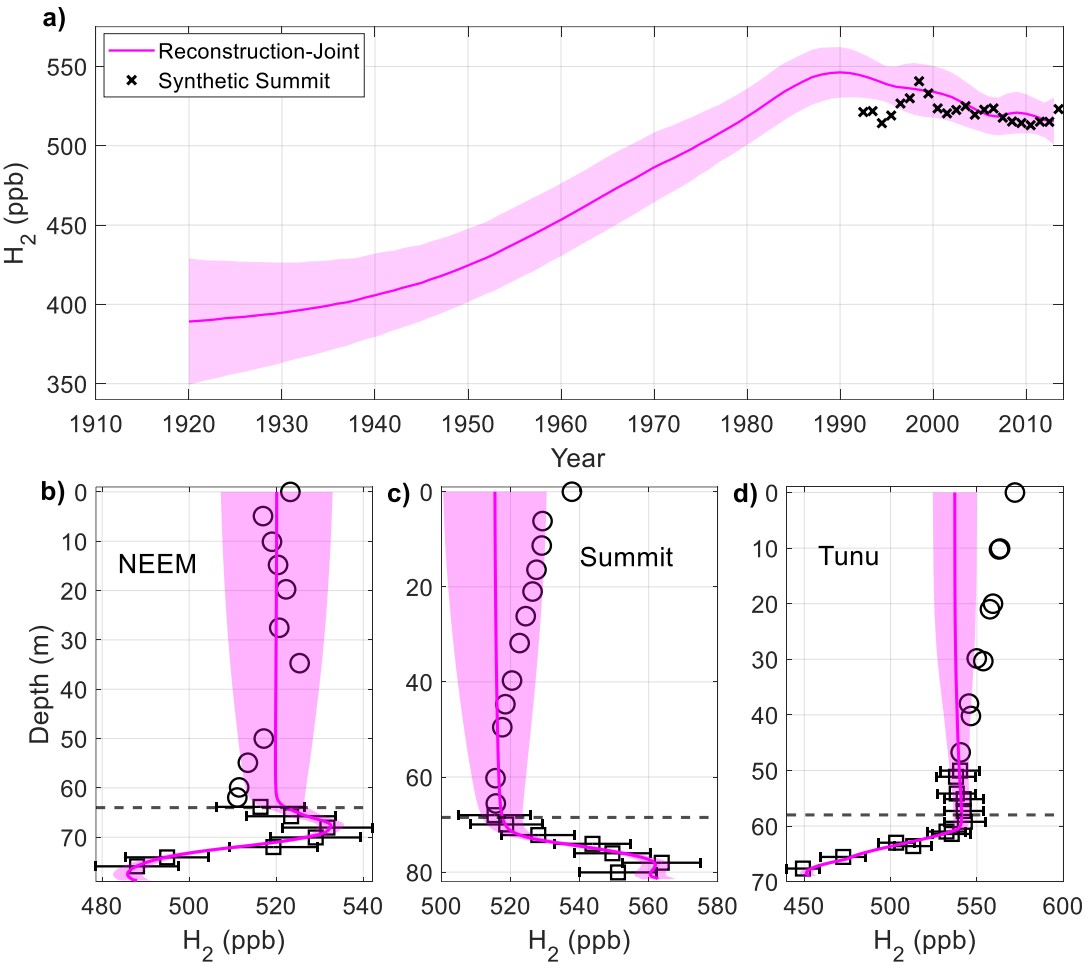

## 5. Comparison of firn air reconstruction with atmospheric flask measurements

The firn air reconstructions are compared to available atmospheric $H_2$ flask measurements from the high northern
latitudes (Figure 3). Direct comparisons between the firn reconstructions and atmospheric measurements at other sites are
complicated by spatial variability in $H_2$ levels in the high northern latitudes. The best temporal coverage for flask
measurements is from the Barrow, Alaska and Alert, Nunavut sampling sites. Both sites sample the boundary layer and are
influenced by their proximity to seasonally active soils. $H_2$ levels over Greenland are more representative of the free
troposphere due to higher elevation and remoteness from active soils. To make this comparison, we utilize NOAA/GML
measurements at Alert, Nunavut and Summit, Greenland from 2010-present (revised to the MPI09 calibration scale; Pétron
et al., 2023) and CSIRO measurements from Alert from 1992 onwards (also on the MPI09 scale).

During the 12 years of NOAA/GML measurements at Summit and Alert, the offset between Summit and Alert
varies from a minimum of 18.9 ppb to a maximum of 29.4 ppb with a mean offset of 25.4 ppb ± 3.0 ppb (1σ) with Summit





levels higher than Alert. We added this offset to the longer CSIRO Alert time series to generate a longer "synthetic Summit"

flask history, assuming that the offset remains constant through time (Figures 1-4, 8). The most conspicuous feature of the
synthetic Summit history is a sharp peak in $H_2$ levels during 1998 which can be attributed to increased biomass burning
emissions during the strong 1997-1998 ENSO event. Firn air cannot resolve such high frequency variability due to diffusive
smoothing, so that feature is absent from our reconstructions. Excepting the 1998 peak, synthetic Summit $H_2$ levels are
largely constant between 525 and 515 from 1992-2013. In contrast, the joint reconstruction shows a trend with $H_2$ levels

decreasing from 545 to 515 ppb over the same time period.

We also examined the atmospheric data for evidence of the $H_2$ maximum around 1990 indicated by the firn air
reconstruction. If this peak had occurred in the atmosphere, we would expect similar trends at all of the high latitude flask
sites despite the differences in absolute $H_2$ levels. Here we examine flask measurements from Barrow, Alaska (Khalil &
Rasmussen, 1990; NOAA/GML) and Alert, Canada (CSIRO; Figure 3). The measurements from NOAA/GML from 1992-

2005 were corrected to the MPI09 calibration scale using the matched flask intercomparison project mentioned previously.
The Khalil & Rasmussen (1990) data are reported on an independent calibration scale which has not been intercompared to
the MPI09 calibration scale. The Barrow data from Khalil & Rasmussen (1990) shows an increase in atmospheric $H_2$ of 2.1
ppb y$^{-1}$ from 1985-1989 and the subsequent NOAA/GML data show a decrease in atmospheric $H_2$ from 1989-1993, at a rate
of 5.0 ppb y$^{-1}$. Together, the trends in these two data sets imply a maximum in atmospheric $H_2$ in 1989, in agreement with the

firn air reconstructions.  The strength of this conclusion is questionable, because the flask intercomparison project did not
include data prior to 1992. From 1993-2010, there is no discernible trend in atmospheric $H_2$ at either Barrow or Alert, while
the firn air reconstructions show a slow decrease in atmospheric $H_2$. The flask measurements suggest a sharper maximum
and more rapid stabilization than the firn air reconstructions.

Firn diffusion imposes smoothing on atmospheric reconstructions.  To test whether the breadth of the atmospheric

$H_2$ peak and slow subsequent decrease in the in the firn air reconstruction could be an artifact of that smoothing, we forced
the firn air model with a synthetic atmospheric history.  The synthetic history was identical to the joint reconstruction before
1989, decreasing by 5 ppb y$^{-1}$ decrease from 1989- 1993 (as observed in the Barrow flask measurements) and subsequently
decreases by 0.5 ppb y$^{-1}$, reaching the same atmospheric concentration as the joint reconstruction in 2013. The depth profiles
generated by the model when forced with the synthetic history show poorer agreement with the measured firn $H_2$ than the

levels generated by the joint reconstruction (Figure S5). This result demonstrates that the firn air measurements are more
consistent with a broad peak and slow decrease in $H_2$ levels than a sharp peak, rapid decrease, and stabilization.

**Figure 3: Comparison of the joint Greenland firn air reconstruction with high northern latitude flask measurements.**
**Magenta line and shading- joint reconstruction as in Figure 2a; purple lines- monthly (solid) and annually-averaged**
**(dashed) flask measurements from Barrow, Alaska made by Khalil & Rasmussen (1990); Blue lines- monthly and**
**annually-averaged flask measurement from Barrow, Alaska made by NOAA/GML (Novelli et al., 1999; Novelli,**
**2006); Red line- annual mean synthetic Summit $H_2$ history (Section 5; Pétron et al., 2023, Langenfelds et al., 2002);**



**Yellow lines- monthly and annually-averaged flask measurement from Alert, Nunavut made by CSIRO (Langenfelds**
**et al., 2002). Measurements from NOAA/GML were empirically corrected to the MPI09 calibration scale using a**
**matched flask inter-comparison project between CSIRO and NOAA/GML.**

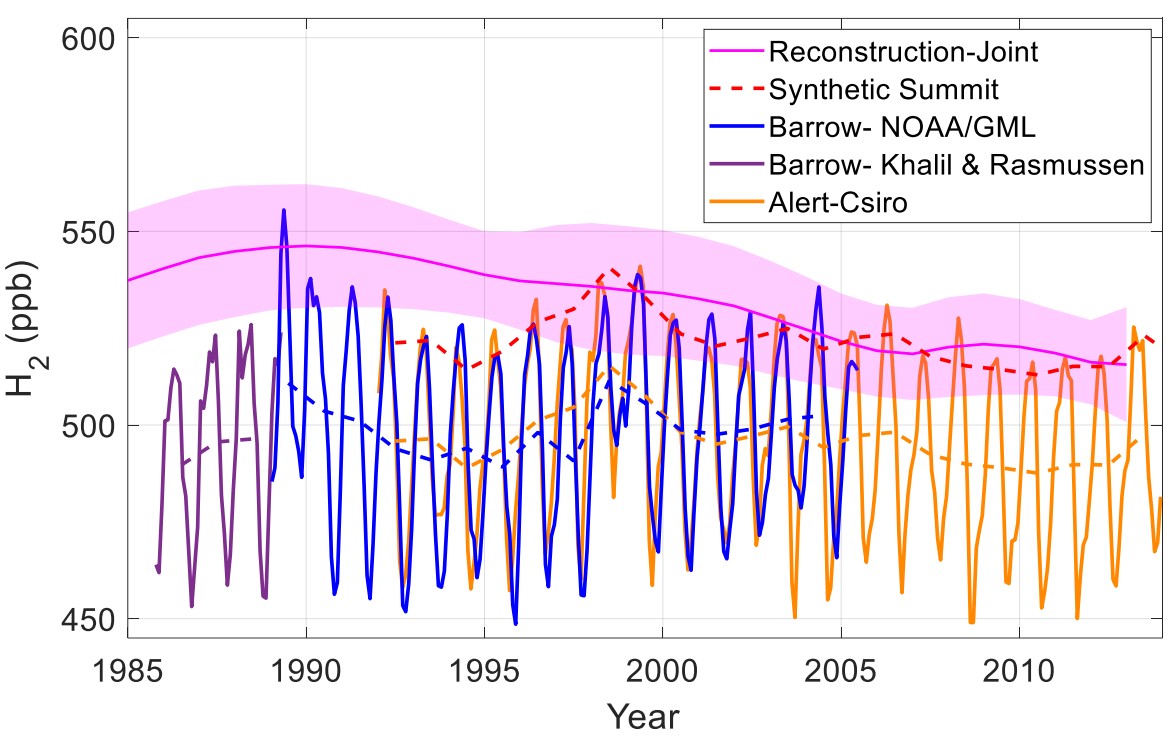



## 6. Comparison to previous firn air reconstructions

395   Petrenko et al. (2013) produced two atmospheric $H_2$ reconstructions from the NEEM firn air data. The reconstructions were generated using two different models: LGGE-GISPA and INSTAAR. Neither of those models include pore close-off fractionation, and Patterson et al. (2021) suggested that maximum in atmospheric $H_2$ inferred from the NEEM firn air could be an artifact of ignoring pore close-off fractionation. Petrenko et al (2013) used the matrix inversion method of Rommelaere et al. (1997), which requires a prescribed smoothing parameter to fully constrain the inversion. Those results

400 were corrected for a calibration revision (see Supplement) and are compared to our joint reconstruction (Figure 4a). The LGGE-GISPA reconstruction shows a rapid rise in $H_2$ levels from 1960-1990, and a rapid decrease from 1990-2008. The INSTAAR reconstruction shows $H_2$ levels rising more gradually from 1960-1985 and decreasing gradually thereafter. The rate of increase of $H_2$ levels prior to 1990 in our joint reconstruction falls between the rates of increase from the LGGE-GISPA and INSTAAR reconstructions. The timing of the atmospheric peak in our joint reconstruction shows good

405 agreement with the timing from the LGGE-GISPA reconstruction, but the magnitude of the peak is in better agreement with the INSTAAR reconstruction. The rate of the decline in atmospheric $H_2$ from 1990-2008 from our joint reconstruction is nearly identical to the rate from the INSTAAR reconstruction. The late 20[th] century maximum in atmospheric $H_2$ is a robust feature of all Greenland firn air reconstructions. According to our model, pore close-off induced enrichment is <1% at the depth of the observed $H_2$ maximum in the lock-in zone. That is, the observed lock-in maximum is not caused by enrichment,

410 but by a historical atmospheric maximum.

   We conducted a similar joint reconstruction for $H_2$ firn measurements from two Antarctic sites: South Pole and Megadunes. The results of that joint reconstruction are compared to previously published reconstructions from those individual sites from (Patterson et al., 2020; 2021; Figure 4b). The differences between the joint reconstructions and the previously published reconstructions are relatively minor. The joint reconstruction shows a slightly slower increase in $H_2$

415 levels prior to 1965 and a more rapid increase from 1965-1980. After 1980, the joint reconstruction is in good agreement with the previously published reconstructions. Overall, the earlier Antarctic reconstructions exhibit a more uniform rate of increase over the past century. This is likely due to the use of an arbitrary smoothing parameter in the earlier studies, while an autocorrelation function was used in the joint reconstruction.




**Figure 4: Comparison of our joint reconstructions with previously published firn air reconstructions a) Magenta line and shading- joint Greenland reconstruction as in Figure 2a; purple lines- Reconstructions from Petrenko et al. (2013) using the LGGE-GISPA model (dotted line) and INSTAAR model (dashed line); black x's- annual mean**

**synthetic Summit $H_2$ history (Section 5; Pétron et al., 2023, Langenfelds et al., 2002). b) Blue line and shading- joint Antarctic reconstruction and associated ±1σ uncertainty; dashed maroon line- South Pole firn air reconstruction from Patterson et al., 2020; dashed red line- Megadunes firn air reconstruction from Patterson et al., 2021; black x's- $H_2$ annual means from measurements made by NOAA/GML at Palmer, Syowa, South Pole, and Halley Stations, Antarctica and Cape Grim Observatory, Tasmania (Novelli et al., 1999; Novelli, 2006); black circles- $H_2$ annual**

**means from measurements made by CSIRO at Casey, South Pole, and Mawson Stations, Antarctica and Cape Grim and Macquarie Island, Tasmania (Langenfelds et al., 2002); $H_2$ annual means from measurements made by Khalil & Rasmussen (1990) at Palmer Station, Antarctica, and Cape Grim, Tasmania**

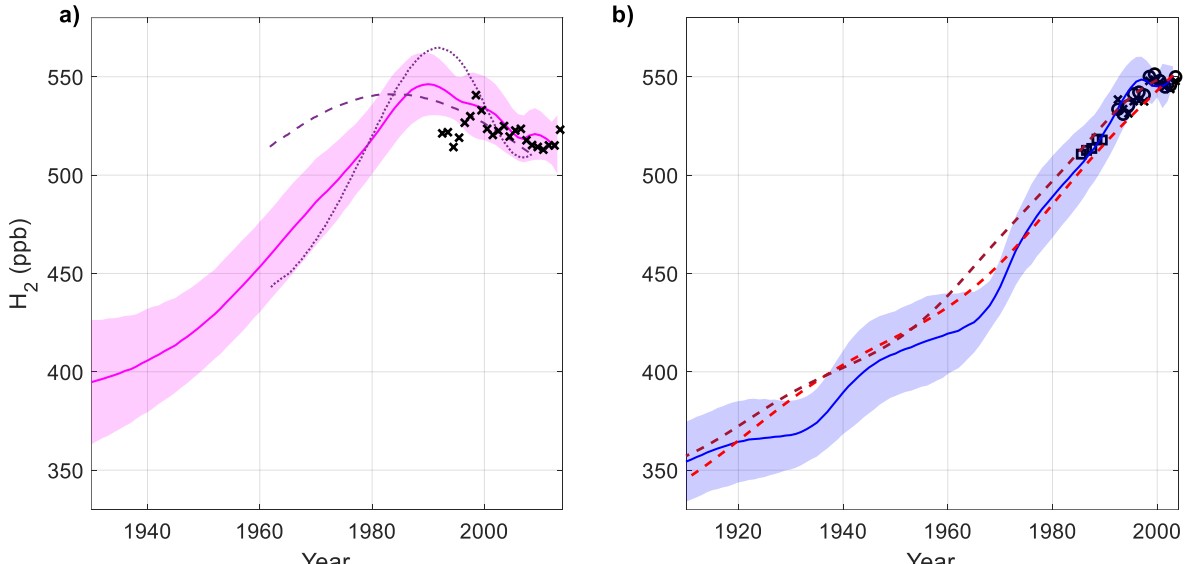

## 7. Investigating the impact of bias in the pore close-off fractionation parameterization

Ne and $H_2$ have similar permeability in ice due to their similar molecular diameters (Patterson & Saltzman, 2021; Satoh et al., 1996). Atmospheric Ne is constant, making it a useful tracer for assessing the effects of pore close-off fractionation. The UCI_2 firn air model underestimates pore close-off induced Ne enrichment at the base of the firn at NEEM and Summit. If a firn air model underestimates Ne enrichment, it is highly likely to also underestimate pore close-off enrichment for $H_2$. An underestimate of pore close-off induced enrichment would yield a reconstruction that is biased high,

and a high bias could be responsible for the reconstructed 1990's peak in $H_2$ that is not observed in the surface air flask network measurements and for the differing trends in the reconstruction and the surface air flask network after the peak. Additionally, the bias could account for the unexpectedly high reconstructed Greenland $H_2$ levels relative to reconstructed Antarctic levels. We consider bias in the Greenland record to be more likely than bias in the Antarctic record because there is good model-measurement agreement for Ne enrichment at the two Antarctic ice core sites. Additionally, the individual





Antarctic reconstructions are in good agreement with each other and with high latitude Southern Hemisphere atmospheric flask measurements since the late 1980's (Figure 4b; Patterson et al., 2020; 2021).

Pore close-off fractionation is expected to substantially enrich $H_2$ and Ne levels in base of the firn (Section 3.1). $\delta^{22}Ne/N_2$ at the base of the firn was measured as 179‰ at NEEM and 112‰ at Summit where $\delta^{22}Ne/N_2$ of a sample is defined based on the relative mixing ratios of $^{22}Ne$ and $N_2$:


$$(\delta^{22}Ne/N_2)_{sample} = [(^{22}Ne/N_2)_{sample}/(^{22}Ne/N_2)_{atmosphere} - 1] * 1000 \qquad (11)$$

The UCI_2 firn air model generates enrichments of only 122‰ at NEEM and 90‰ at Summit at the base of the firn, significantly lower than the measurements (Figure 5). The Ne content of the firn air at Tunu has not been measured. As a
first attempt to improve agreement between measured and modeled Ne enrichment, we re-tuned the closed porosity profiles for NEEM and Summit to optimize model-measurement agreement for Ne enrichment. However, the optimization yielded closed porosity profiles that are qualitatively different from previously published parameterizations and are probably unrealistic (e.g. Schwander et al., 1989; Goujon et al., 2003). Therefore, the parameterization of some physical process is likely incorrect.  Possible candidates include the rate of densification, the rate of bubble close-off, and the rate of
pressurization of bubbles.  A detailed observation-based investigation of these processes at these sites would require field data that does not currently exist. Instead, we explored how altering the model parameterizations of these processes could improve the model-measurement agreement for Ne and examined the sensitivity of the firn air reconstruction to various physical assumptions.










**Figure 5: Measured and modelled $\delta^{22}Ne/N_2$ at NEEM (a) and Summit (b). Magenta lines- $\delta^{22}Ne/N_2$ generated by the
original UCI_2 model; dashed lavender lines- $\delta^{22}Ne/N_2$ generated by the model with seasonal layering implemented
(Section 7.1); maroon lines- $\delta^{22}Ne/N_2$ generated by the model with reduced bubble compression (Section 7.2); dotted
cyan lines- $\delta^{22}Ne/N_2$ generated by the model with both seasonal layering and reduced bubble compression (Section
7.3); black squares- $\delta^{22}Ne/N_2$ measurements corrected for gravitational fractionation and depth-averaged.**

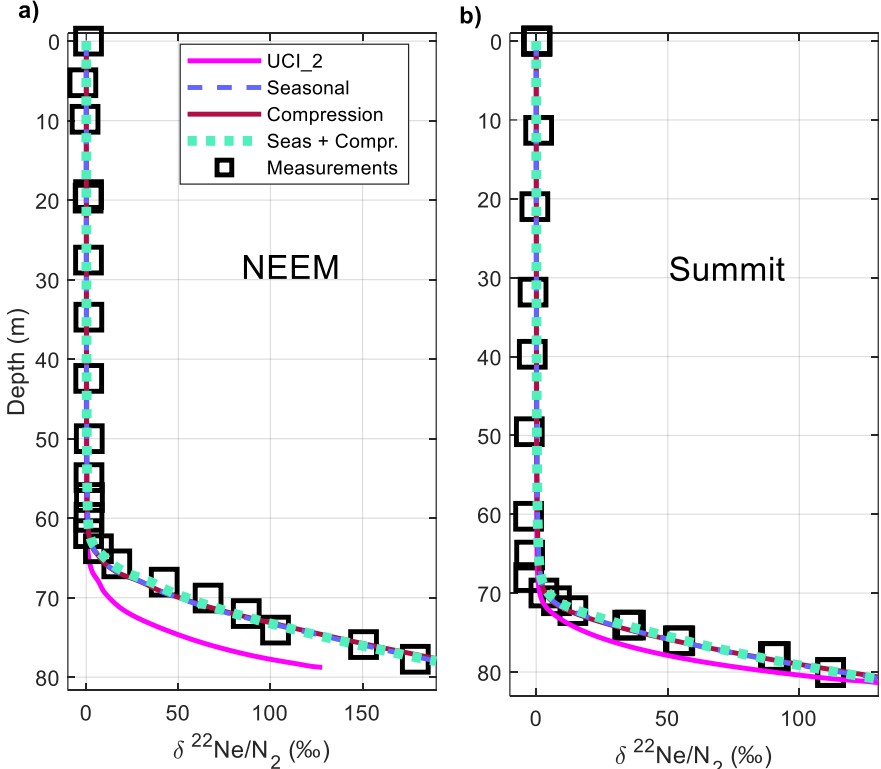


### 7.1 Method 1: Seasonal layering

Here we explored whether incorporating seasonal differences in densification could yield improved model-
measurement agreement for Ne. Winter layers at both NEEM and Summit densify more rapidly than summer layers
(Martinerie et al., 1992; Fujita et al., 2014). Fujita et al. (2014) attribute increased winter densification and deformation to
seasonal differences in ion concentrations in the snow. Fujita et al. (2014) further show that bubbles occlude and pressurize
preferentially in the dense winter layers before the summer layers. The UCI_2 model (and most firn air models) utilizes
seasonally averaged bulk density and porosity to calculate bubble close-off and pressurization. This should lead to errors
because of the non-linearities in the parameterization. The importance of layering in the firn to bubble trapping has been
discussed extensively by Mitchell et al. (2015) and Fourteau et al. (2017).





In this experiment, we divide each model layer into a winter and summer "sub-layer." We arbitrarily specify that half of the mass of each layer is winter accumulation and half is summer accumulation. The density of each sub-layer is estimated based on Fujita et al., (2014; Figure S6), and the total porosity of the sub-layers is calculated using equation 1. The bulk density of the firn is not changed from the original model. The original bulk closed porosity profile does not yield agreement between measured and modeled Ne enrichment, so the closed porosity profile is revised.  The closed porosity of

each sub layer ($s_c$) is prescribed as a function of the total porosity and density of that sub layer:

$$s_c = \frac{s_{total}}{1+\exp\left(-m(\rho_{firn}-\rho_0)\right)} \tag{12}$$

Where $m$ and $\rho_0$ are free parameters that are tuned to yield agreement between measured and modeled Ne enrichment (Table

3; Figures 5 and 7). It has been demonstrated that winter and summer layers have similar close-off characteristics as a function of density, so the tuneable parameters are the same for both winter and summer sub-layers (Martinerie et al., 1992; Fourteau et al. 2017).  Bubble close-off and pressurization is calculated independently for winter and summer sub-layers using the same parameterization as in the UCI_2 model after Severinghaus & Battle (2006; equations A11-A13). Then, the bulk closed porosity and bubble pressure of each layer is determined from its constituent sub-layers, and the model is run as

before using the calculated bulk properties. New $H_2$ age distributions are calculated, and the new age distributions are used to generate a revised joint reconstruction (Section 3.2; Figures 6 and 8).

### 7.2 Method 2:  Reduced bubble compression

The bubble pressurization scheme used in the UCI_2 model assumes that closed bubbles compress at the same rate

as the total porosity after Severinghaus & Battle (2006). Bubble trapping models that assume iso-compression have been shown to overestimate total air content in bubbles at the Lock-in and Vostok sites in East Antarctica (Fourteau et al., 2017). In that work, better model-measurement agreement was achieved by reducing the compressibility of closed bubbles by 50%. Here, we explored whether reducing the compressibility of bubbles could yield improved model-measurement agreement for Ne enrichment. We modified equation A12 in Severinghaus & Battle (2006), which calculates the incremental new volume

of air at ambient pressure ($\Delta V_{b(i)}$) occluded in bubbles between layer ($i$-$1$) and layer ($i$) per unit volume of firn:

$$\Delta V_{b(i)} = \frac{2\left\{s_{c(i)}-s_{c(i-1)}\left[(1-k_{comp})+k_{comp}\frac{s_{total(i)}}{s_{total(i-1)}}+\frac{\rho_{firn(i)}}{\rho_{firn(i-1)}}-1\right]\right\}}{(1-k_{comp})+k_{comp}\frac{s_{total(i)}}{s_{total(i-1)}}+1} \tag{13}$$

Where $k_{comp}$ is a dimensionless free parameter that represents the degree of compressibility of the bubbles. In the limit of

$k_{comp}=1$, equation 13 reduces to equation A12 of Severinghaus & Battle (2006), and bubbles compress at the same rate as the open pores. In the limit of $k_{comp}=0$, the bubbles are completely incompressible. Again, the original closed porosity profile





does not yield agreement with the measured Ne enrichment, so a new closed porosity profile is prescribed using equation 12. The three tuneable parameters from equation 12 and 13 ($k_{comp}$, $m$, and $\rho_0$) are simultaneously optimized to give the best fit between modeled and measured Ne enrichment (Table 3; Figures 5 and 7). Then, the model is used to calculate new age
distributions for $H_2$, which are used for new reconstructions (Figures 6 and 8).

### 7.3 Method 3: Seasonal layering and reduced bubble compression

Lastly, we incorporated both seasonal layering and bubbles with reduced compressibility into the UCI_2 model. The density of the summer and winter layers is identical to Method 1. Instead of equation A12 from Severinghaus & Battle
(2006), we use equation 13 to calculate bubble compression. Again, $k_{comp}$, $m$, and $\rho_0$ are tuned simultaneously to optimize the fit between measured and modeled Ne enrichment (Table 3; Figures 5 and 7). The model is run for $H_2$, and the new age distributions are used to generate new $H_2$ reconstructions (Figures 6 and 8). We note that all three new parameterizations have a negligible effect on the modelling of trace gases like $CO_2$ and $CH_4$, which are not affected by pore close-off fractionation (Figure S7)

**Table 3**- Optimized tunable parameters for each method described in 7.1-7.3

| Site | Method 1 | | | Method 2 | | | Method 3 | | |
|------|----------|---|---|----------|---|---|----------|---|---|
| | $\rho_0$ $(kg\ m^{-3})$ | $m$ $(m^3\ kg^{-1})$ | $k_{comp}$ | $\rho_0$ $(kg\ m^{-3})$ | $m$ $(m^3\ kg^{-1})$ | $k_{comp}$ | $\rho_0$ $(kg\ m^{-3})$ | $m$ $(m^3\ kg^{-1})$ | $k_{comp}$ |
| NEEM | 827 | 0.070 | 1.00 | 815 | 0.070 | 0.75 | 812 | 0.138 | 0.69 |
| Summit | 832 | 0.070 | 1.00 | 821 | 0.070 | 0.75 | 818 | 0.138 | 0.69 |


### 7.4 Revised reconstructions

The reconstructions generated by the three methods are very similar (Figure 8). The most significant consequence of increasing pore close-off fractionation is a reduction in the atmospheric maximum around 1990 and a more modest decrease in atmospheric levels from 1990-2005. That is, higher modeled enrichment due to pore close-off fractionation
yields correspondingly lower reconstructed $H_2$ levels. The 1990 maximum in the revised reconstruction is ~530 ppb, compared to 547 ppb in the original. After 1990, $H_2$ levels in the revised reconstructions decrease very slowly to 515 ppb in 2013, converging with the original reconstruction. The modest decrease in $H_2$ levels during the 1990's in the revised reconstructions is in better agreement with high northern latitude flask measurements and synthetic Summit history, which do not show a discernible trend in atmospheric $H_2$ during the 1990's.

Because the older firn air that was sampled at Tunu is excluded from this analysis, the reconstructions are only meaningful after about 1950. Overall, $H_2$ levels in the revised reconstructions are reduced compared to the original





reconstruction by 15-30 ppb from 1950-1990 (Figure 8). The new reconstructions remain 0-15 ppb higher than the Antarctic reconstructions, so the unexpectedly high reconstructed Greenland levels relative to Antarctic levels can be only partially explained by the underestimate of pore close-off induced enrichment (Figure 9).

We can draw several important conclusions from these results:  1) $H_2$ atmospheric reconstructions are sensitive to pore close-off fractionation, 2) the large 1990's peak in the Greenland reconstructions is likely an artefact of underestimated pore close-off fractionation in the UCI_2 model, 3) as long as sufficient fractionation is achieved, the reconstructions are insensitive to the specific parameterization and internal model physics. All three of the approaches resulted in an increase in the integrated area of the age distributions and a slightly older shift in mean age (Figure 6). All three methods greatly

improve the fit between modelled and measured Ne enrichment at both sites. The similarity in optimized tuneable parameters suggest that a common mechanism underlies the UCI_2 model's underestimate of pore close-off induced enrichment at both NEEM and Summit (Table 3). It is likely that the common mechanism is associated with bubble close-off and pressurization. Further investigation of layering and bubble close-off using high resolution measurements of density, porosity, and air content, in conjunction with measurements of highly diffusive trace gases like Ne or He would be useful in constraining pore

close-off fractionation for future Greenland firn air work.











**Figure 6: Open porosity H₂ age distributions resulting from several approaches to approximating pore close-off fractionation for H₂ at a depth of 75.9 m at NEEM (a) and at a depth of 80.1 m at Summit (b). Magenta lines- original UCI_2 model; dashed lavender lines, maroon lines, and dotted cyan lines are from the model with seasonal layering implemented (Section 7.1), with reduced bubble compression (Section 7.2), and with both seasonal layering and**

**reduced bubble compression (Section 7.3) respectively.**

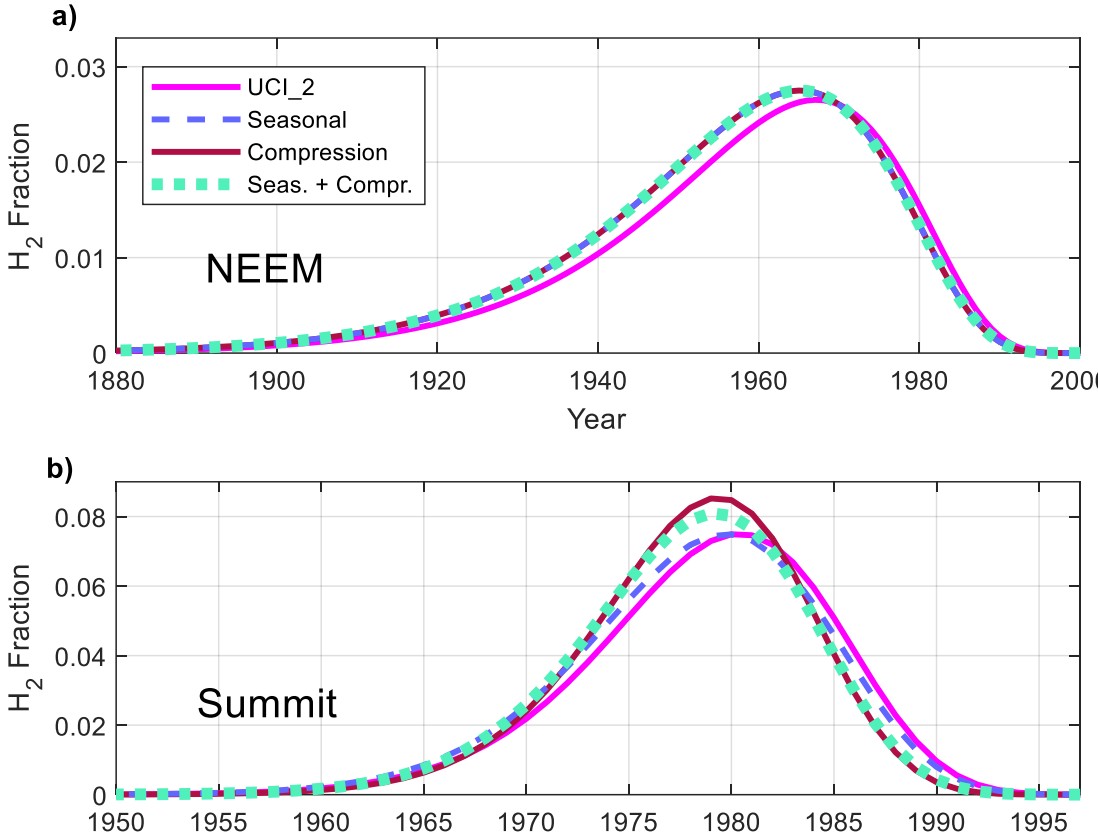








**Figure 7: Model closed porosity profiles used for NEEM (a) and Summit (b). Magenta lines- original closed porosity profile used in the UCI_2 model; dashed lavender lines, maroon lines, and dotted cyan lines are retuned closed porosity profiles used for the model with seasonal layering implemented (Section 7.1), with reduced bubble compression (Section 7.2), and with both seasonal layering and reduced bubble compression (Section 7.3)**
**respectively.**

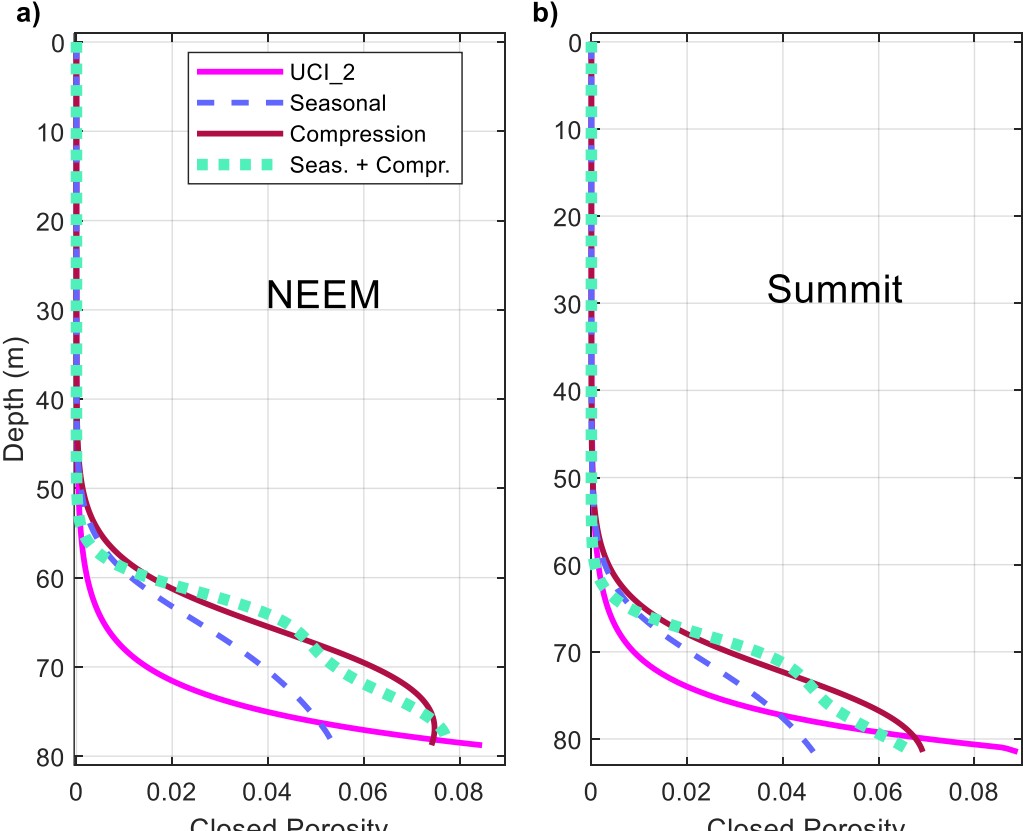




**Figure 8: Reconstructions of atmospheric H₂ levels from Greenland firn air. Magenta line- joint reconstruction as in**
**Figure 2a; dashed lavender lines, maroon lines, and dotted cyan lines are joint reconstructions calculated from the**
**model with seasonal layering implemented (Section 7.1), with reduced bubble compression (Section 7.2), and with**
**both seasonal layering and reduced bubble compression (Section 7.3) respectively; black x's annual mean synthetic**
**Summit H₂ history (Section 5; Pétron et al., 2023, Langenfelds et al., 2002). Grey shading is the approximate ±1σ**
**uncertainty band for the three corrected reconstructions. The upper bound is the maximum of the sum of the three**
**histories and their respective 1σ uncertainties for each year, and lower bound is the minimum of the difference**
**between the three histories and their respective 1σ uncertainties.**

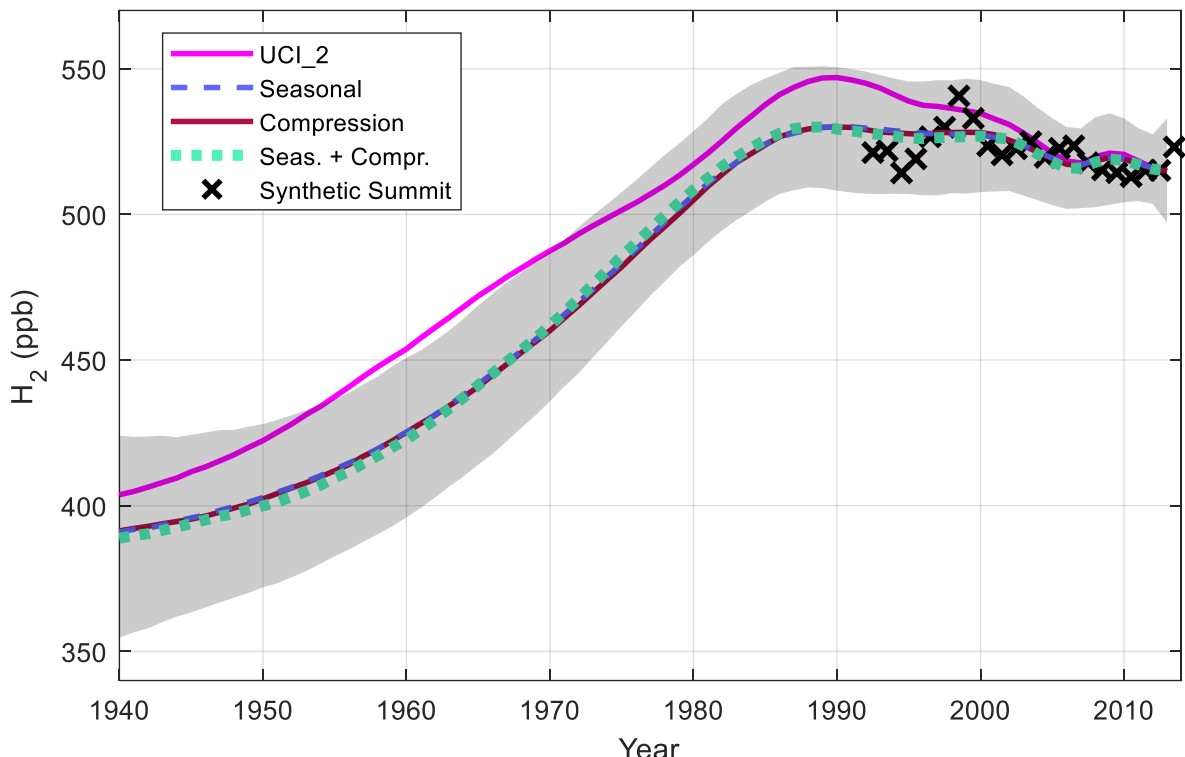


## 8. *In situ* production

It is possible that some of the differences between the independent reconstructions (Figure 1) could be caused by *in situ* production, but little is known about in situ H₂ production in snow and ice. It is useful to consider evidence for *in situ* production of CO because both trace gases may be photochemically produced from carbonyl-containing precursors. CO has
been shown to be produced photochemically from CH₂O in surface snow (Hahn et al., 2001). At Devon Island, Canada CO levels in the firn increased steadily with depth in both the diffusive and lock-in zones, exhibiting clear evidence of in situ production (Clark et al. 2007). By contrast, NEEM and Summit firn air both exhibit CO maxima in the firn air column





associated with a late 20<sup>th</sup> century atmospheric maximum, with CO levels decreasing with increasing depth below (Petrenko et al., 2013). Petrenko et al. (2013) concluded in situ production at those sites was not sufficient to bias the atmospheric

reconstruction. Those researchers used three lines of evidence to support their conclusion. First, the ice age at the depth of the observed lock-in zone CO peak is different at the two sites, so the peak cannot be attributed to a widely deposited layer of organic impurities. Second, the $CO/CH_4$ ratio at a given depth is in reasonable agreement at NEEM and Summit. CO and $CH_4$ have similar gas-phase diffusivities, so the mean age of the two gases at a given depth is comparable. $CH_4$ is known to be well preserved in Greenland firn air. If *in situ* production differentially affected CO levels at the two sites, it would be

expected that $CO/CH_4$ ratios would diverge. Third, forcing a firn air model with an atmospheric flask derived CO history yields good agreement with the measured depth profile down to the top of lock-in zone where the firn air is too old to be constrained by flask measurements.

Greenland ice cores do exhibit evidence of in situ CO production. Early discrete measurements on Eurocore ice showed low, steady preindustrial CO levels (Hahn and Raynaud, 1998; Hahn et al., 1996). By contrast, Fain et al. (2022)

reported higher and more variable CO levels from several Greenland sites (including NEEM and Tunu) using continuous flow and discrete methods. The elevated CO levels were apparently associated with localized organic impurities in the ice. The high variability in CO detected by the continuous flow method is not compatible with continuous production in the firn, where such signals should be smoothed by diffusion. It is therefore likely that the high frequency variability observed by Fain et al. (2022) originated below the lock-in zone. However, the ice core data do not preclude the possibility of some in

situ production having occurred in the overlying firn.

The production of CO in Greenland firn air and ice is not well understood, and no physical mechanism has been proposed that can explain all of the observations. If the production is photochemical, one would expect it to be much faster in the firn than in ice, due to the much larger actinic flux. Until the mechanism is established, it is difficult to assess whether and to what extent *in situ* production might influence firn air $H_2$. Depending on the depth and mechanism, production of $H_2$

in the firn column would cause a high bias during some or all of our atmospheric reconstructions. Further research on this issue is needed.

## 9. Summary and conclusions

In this study we use all available firn air measurements of $H_2$ at Greenland and Antarctic sites to reconstruct

atmospheric $H_2$ levels over the past century. The joint Antarctic reconstruction covers the 1900-2003 period and is based on firn air measurements from South Pole and Megadunes (Patterson et al., 2020; 2021). That reconstruction shows good site-to-site agreement and good agreement with the atmospheric flask network since the late 1980's. The good fit between measured and simulated Ne enrichment obtained using the UCI_2 firn air model gives us high confidence in the modelling of pore close-off fractionation at the Antarctic sites.

Reconstructing $H_2$ levels over Greenland is more challenging because of inter-site biases and because the standard UCI_2 firn air model underestimates the enrichment of Ne at NEEM and Summit. It is likely that the model also





underestimates pore close-off induced enrichment of $H_2$ because of the similar permeability of the two gases in ice (Patterson & Saltzman, 2021; Satoh et al.1996). Three alternative parameterizations of pore close-off fractionation were explored to assess the sensitivity of the reconstructions to an underestimate of pore close-off induced enrichment. These revised reconstructions all show a less prominent atmospheric maximum and more stable $H_2$ levels during the late 20th century, in better agreement with the atmospheric flask measurements.

There is clearly uncertainty regarding the modelling of the physical processes influencing pore close-off fractionation, particularly in Greenland firn air. A path to increasing confidence in firn air reconstructions for $H_2$ is to obtain new observations of firn air chemistry with high resolution measurements of firn density, porosity, and bubble total air content. Absent additional measurements, we believe that the joint reconstructions presented here are the best estimate for the evolution (and uncertainty) of atmospheric $H_2$ during the 20th century over Antarctica and Greenland (Figure 9). Ice core records at high accumulation sites should also be collected in order to constrain $H_2$ levels over the late pre-industrial and industrial eras. Such records will be important to better understanding the global $H_2$ cycle, the impact of human activities on $H_2$ in the past, and the atmospheric response to increasing anthropogenic $H_2$ emissions in a warming climate.

One surprising result is that our reconstructions show higher $H_2$ levels over Greenland than over Antarctica prior to 1990. This implies a reversal in the sign of the interhemispheric difference during the late 20th century. It is believed that Northern hemispheric $H_2$ levels are lower than Southern hemisphere levels today due to strong uptake of atmospheric $H_2$ by Northern hemisphere soils. The N/S interhemispheric difference in atmospheric $H_2$ occurs in spite of the fact that direct anthropogenic emissions of $H_2$ occur predominantly in the northern hemisphere. The dominance of northern hemispheric soil sink is largely a function of the asymmetry in land area between the two hemispheres. While the soil sink may vary over time due to changes in ecosystems, land use, or climate, a dramatic change would be required in order to have caused a reversal of the interhemispheric difference. In this respect, the firn air reconstructions present a challenge to our understanding of the global atmospheric $H_2$ budget and careful examination of our conclusion that the northern hemisphere firn air reconstruction is not biased by *in situ* $H_2$ production in Greenland firn.



**Figure 9: Best estimates for the atmospheric history of H₂ in Greenland and Antarctica over the past century. Blue line- joint Antarctic firn air reconstruction as in Figure 4b; Green line- joint Greenland firn air reconstruction obtained from the mean of the three reconstructions using revised pore close-off parameterizations (Seasonal, Compression, and Seasonal+Compression; shown in Figure 8). Shaded areas are ±1σ uncertainties.**

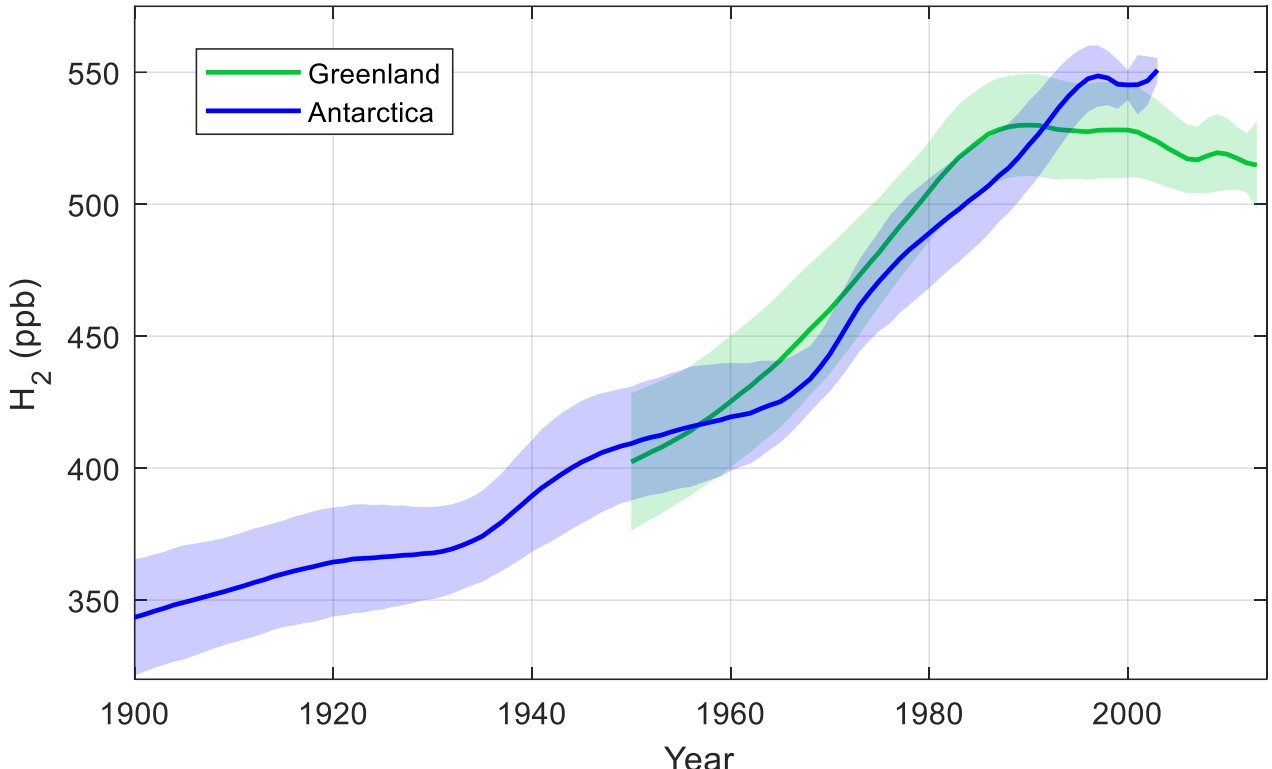

## Data and code availability

The data and code used in this research has been submitted to the DRYAD data repository, and we will provide DOI identifiers before the end of the review process.

## Author contributions

J.D.P. carried out the model simulations and data analysis; M.A. contributed to data analysis; A.M.C., G.P., R.L.L, and P.B.K. made calibration corrections and contributed atmospheric flask measurements; J.P.S. collected the field samples, measured δ²²Ne/N₂, and assisted with firn air modeling; V.V.P collected field samples and contributed to firn air measurements, E.S.S. contributed to modeling, data analysis, and editing the manuscript; and J.D.P. wrote the manuscript.



## Competing interests

The authors declare that they have no conflict of interest.

## Acknowledgements

The authors would like to thank Christo Buizert for helpful discussions about firn air modelling. This research was supported by the NSF (OPP-1907974 and OPP-2019719).

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
