# Peer review of "Reconstructing atmospheric H2 over the past century from bi-polar firn air records"

_Climate of the Past, 2023_

## Author Comment (AC2)

Reviewer 2- Christo Buizert

Patterson and co-authors present new firn air-based atmospheric reconstructions of atmospheric H2 over the last century, combining data from several sites and both hemispheres. The difficulty in reconstructing atmospheric H2 from firn and ice core records is the molecular size-dependent bubble close-off fractionation. This fractionation results in H2 depleting in mature ice, and a corresponding enrichment in the firn air. The authors present several ways of correcting for this firn artifact.

I find their reconstructions convincing and worthy of publication in Climate of the Past. The work seems technically sound, and has been described clearly. I have a few suggestions I would like the authors to consider, in particular regarding the structure of the manuscript.

We thank the reviewer for his supportive comments, thorough review, and detailed suggestions. Specific comments are addressed individually below.

(1), I wonder why the authors chose to present the "regular" UCI_2 inversions as their main result (Figs 1-4), and then the various scenarios with altered pore close-off schemes as some sort of alternate or special case (Figs. 5-9). In my mind, one cannot reconstruct atmospheric H2 meaningfully without getting the close-off size fractionation correctly, and therefore I think the "altered" scenarios are far more convincing. Case in point, these "altered" scenarios provide a good fit to the atmospheric flask observations that are the gold standard in atmospheric trace gas reconstruction.

If this were my paper, I would not bother with the regular UCI_2 model, as it is clear it cannot fit the Neon data nor the direct atmospheric flask data. It is customary to calibrate (or tune) the diffusivity profile to trace gas records; in this situation I would consider fitting the Neon data as part of the model calibration (tuning). I would have only presented the "altered" scenarios. This would also result in a much shorter and more focused manuscript. I suspect the authors trust the altered scenarios better themselves, as this is what they plot in their last summary figure (Fig. 9). Perhaps this review can be a justification for the authors to make this simplification of the manuscript. On the other hand, I realize this is a big revision I am suggesting, and I would still be supportive of publishing this paper if the authors decide to keep the current structure.

At the very least the authors should communicate more clearly which inversion they believe to be most realistic, so that users of these reconstructions know which one to plot. I would advocate strongly that the most realistic ones are those that can fit the dNe/N2.

We agree with the reviewer that the "altered" scenarios are more realistic and that this should be clearly communicated in the paper. We prefer to include the UCI_2 model results in the manuscript as a "baseline," in order to contextualize need for altered scenarios, particularly since the altered scenarios are essentially *ad hoc*. Including the UCI_2 model results also demonstrates the dramatic differences between the Greenland sites and the Antarctic sites.

Revisions related to this point:

L751-753: Absent additional measurements, we believe that the results presented in Figure 9 are the best estimate for the evolution (and uncertainty) of atmospheric H2 during the 20th century over Antarctica and Greenland.

L139-154: The UCI_2 firn air model is a 1-dimensional finite-difference advective-diffusive model that is used to simulate the evolution of trace gas levels in firn air. The UCI_2 model has been used to successfully analyse H2 ¬levels in firn air at two Antarctic sites, including the effects of pore close-off fractionation (see below; Patterson et al., 2020; 2021). As discussed in Section 7, the default model parameterizations do not adequately capture the effects of pore close-off fractionation at two of the Greenland sites; nonetheless, the model provides a useful baseline for analysing atmospheric H2 levels in firn air.

L746-748: Additionally, the alternative parameterizations of pore close-off fractionation yield better model-measurement agreement for δ22Ne/N2. The improved agreement indicates that the reconstructions generated by the alternative parameterizations are more realistic than the reconstruction generated by the UCI_2 model.

(2) The authors apply a gravitational fractionation correction to the data via the d15N data. It would be trivial to similarly apply a close-off fractionation to the data via the dNe/N2 data (assuming that the dH2/N2 fractionation is the same as the dNe/N2, which the authors assume already). In this way the bubble close-off would be included empirically, and would not have to be modeled. Can you add an inversion using such an empirical data correction? My prediction would be that it matches the "altered" solutions.

The inversion described above is not identical to the alternative solutions. This is due to the small difference in the age distribution of $H_2$ caused by mixing between the closed bubbles and open pores. As a result, explicit modeling of pore close-off fractionation is necessary. We have added a Figure to the supplement demonstrating this (Figure S7), and made the following addition:

L495-497:  Because mixing between the closed bubbles and open pores affects the modeled H2 age distributions, measurements cannot be empirically corrected for pore close-off fractionation as they are for gravitational fractionation (equation 2; Figure S7). Instead, pore close-off fractionation must be explicitly modeled.

(3) The discussion around the close off fractionation (section 7) is not as clear as it could have been. A few suggestions:

(3a) Can you add a panel to Fig. 7 showing the bubble pressure in the model? That is ultimately what drives the fractionation, so it is critical to have this information.

We like the reviewer's suggestion to show the physical factors responsible for fractionation on Fig. 7.  At equilibrium, modelled pore close-off fractionation is controlled primarily by the ratio of the volume-weighted average pressure in the open and closed pores to the ambient pressure, adjusted for a small amount of mixing. We decided to plot this ratio (R) rather than bubble pressure alone, making it more obvious for readers why the three alternative parameterizations yield the same neon enrichment. The text was also revised as follows:

L468-481: At equilibrium, modelled pore close-off fractionation is controlled primarily by the ratio of the volume-weighted average pressure in the open and closed pores to the ambient pressure, adjusted for mixing. We introduce a new parameter, *R*:

$$R = (P_{bubble}s_c/s_{total} + P_{ambient}s_o/s_{total})/P_{ambient} \qquad (12)$$

In the UCI_2 model, $R$ begins to increase too deep in the firn to capture the shallower $\delta^{22}Ne/N_2$ measurements (Figure 5; Figure 7). As a first attempt to improve agreement between measured and modeled Ne enrichment, we re-tuned the closed porosity profiles for NEEM and Summit to optimize model-measurement agreement for Ne enrichment using previously published parameterizations, including Schwander et al. (1989), Goujon et al. (2003), and Mitchell et al. (2015). However, in all cases, the optimization yielded closed porosity profiles that are qualitatively different from previously published estimates and probably unrealistic. When the optimizations were constrained to generate realistic closed porosity profiles, in all cases, we found that $R$ increases too rapidly with depth capture the $\delta^{22}Ne/N_2$ at both the top and the bottom of the lock-in zone. Therefore, in order to generate the necessary $R$ profile while maintaining a realistic closed porosity profile, some other physical process must be modified in addition to the closed porosity profile.

(3b) I don't understand why the "reduced compression" scenario would result in more dNe/N2 fractionation. That makes no sense physically, as halving the pressure should also half the dNe/N2 anomaly. From Fig. 7a I suspect that the authors also altered the closed porosity parameterization, and that this is what drives the enhanced dNe/N2. Please check/confirm.

The reviewer is correct. The closed porosity profile was changed in addition to the compression. We have tried to clarify that in all three scenarios, the closed porosity parameterization is changed in addition to some other physical process (see response to 3a). In addition, L571-572 now states: "Again, the original closed porosity profile does not yield agreement with the measured Ne enrichment, so a new closed porosity profile is prescribed using equation 12. The three tuneable parameters from equations 12 and 13 ($k_{comp}$, $m$, and $\rho_0$) are simultaneously optimized to give the best fit between modeled and measured Ne enrichment (Table 3; Figures 5 and 7)."

(3c) I am not surprised that the regular NEEM closed porosity parameterization gives a poor result for dNe/N2, given that it closes off much too deep (resulting in less pressurization). The Goujon/Martinerie close-off density at that site was artificially increased to match the field observation of the deepest pumping depth. It also ignores layering; including layering automatically results in some shallower trapping that will increase the dNe/N2 anomaly (as shown in Fig. 7a). Have you tried using the porosity parameterization from Mitchell et al. (2015), with the suggested close-off density from that paper? That may solve some of these problems. That parameterization does not produce an abrupt full bubble closure as may be required in some model architectures – this can be added manually perhaps.

We did try the Mitchell parameterization when trying to tune the model to capture the neon enrichment, but the results were not optimal. This is now mentioned in the text (see response to 3a).

(3d) The solutions from all three "altered" scenarios are virtually identical. Could this be because they start trapping bubbles at the same depth (Fig. 7)?

Essentially, yes. More specifically, the three alternative scenarios yield the same neon enrichment because R is essentially identical in all 3 cases (see revised Fig. 7).

(3e) Could you extend the plot in Fig. 7 further down, to for example 90 m? Currently we cannot evaluate how sudden or deep the full bubble close-off occurs (the point where the closed porosity starts to decrease). Sometimes I find it more useful to plot the closed pore fraction, rather than the close porosity itself.

Our model domain does not extend deeper than the depth where closed porosity fraction=1 (see revised Fig. 7). This means that for the UCI_2 model, we have plotted to the point where closed porosity begins to decrease. Because the alternative scenarios are ad hoc, we do not think that extending the domain adds scientific value to the paper. New high resolution measurements of firn density, porosity, and bubble total air content are needed to investigate this question.

(4) Can you elaborate on using a Green's function approach in the presence of bubble close off fractionation? Green's functions assume a linear system response (the sum of two solutions is also a solution to the diffusion equation). Is this true in the presence of close-off fractionation? I suspect it is, but I am not entirely sure. Is the area of under the Green's function greater than 1 in this case?

The reviewer is correct on all accounts. We have tested this by comparing forward model runs to equation 7. We have made the following addition to the manuscript:

L241-243: For most gases, the Green's functions sum to 1 at every depth. In the case of H2 and Ne, the sum of the Green's functions in the lock-in zone is >1 due to pore close-off fractionation.

Other comments:

L33: "second-most"

Done

L41: consider replacing "modern" with "present-day"

Done

L45: Do you have a reference for the OH radical?

The references at the end of the sentence include estimates of the lifetime of H2 with respect to oxidation by OH.  Those papers include information about the OH fields used in their estimates.

L52: Consider also adding a reference to Solomon et al. 2010, who first clearly describe greenhouse forcing from stratospheric H2O

Done

L62: These trends are not very robust, and rely on single year anomalies. Also, how well are Khalil and Rasmussen calibrated with NOAA/GML?

Khalil & Rasmussen data show an increasing trend from 1986-1989 and the NOAA data show a decrease from 1989-1993. Note, there is a more detailed discussion of the flask data in L387-394. We have made the following revisions:

L56-57: Integration of records produced by the different groups has been complicated by calibration issues, which are discussed in greater detail in Section 5.

L61-63: Broadly, the instrumental record shows northern hemispheric H2 levels rising during the late 1980's to a maximum in 1990 and decreasing until 1993.  There is no discernible trend in Northern Hemisphere H2 levels from 1993-2010 (Figure 3).

L79-84: What about Antarctica? Those SH reconstructions are treated as somewhat of an afterthought in this paper, despite the topic being bipolar H2.

This is true… the Antarctic reconstructions are not discussed in as much detail because the differences from our previous work (Patterson et al., 2020; 2021) are small. We have made the following addition to the manuscript:

L81: Additionally, we reanalyse the Antarctic firn air data using a different inversion technique, described in Section 3.2.

Section 2.2: Normally the diffusivity profile is somewhat model-dependent. Can you simply apply the profile from a different model?

We validated the diffusivity profiles by simulating $CO_2$, $CH_4$ (Figure S8). We also ran $SF_6$ and $CH_3CCl_3$. See L193-197.

L134: same depth "were" averaged

Done

L136: intense seasonality: is it much deeper for H2 than for other gases like CO2?

Yes, for two reasons:

1) The seasonality of $H_2$ in the NH is ~10%. For comparison, the seasonality of $CO_2$ is ~4% and the seasonality of $CH_4$ is ~2%

2) The free air diffusivity of $H_2$ is much higher than that of other gases, so seasonal levels penetrate deeper into the firn

L144: different sampling dates?

Text revised to clarify that the differences in profiles reflect both differences in site physical characteristics and changes in atmospheric H2 over the time period during which the various firn air studies were done (17 years):

L140:  The qualitative differences in the depth profiles reflect changes in surface air $H_2$ levels over the period of the firn air studies (1996-2013) and the different physical characteristics of the sites.

L154: This upper 5m is often called the convective zone

We prefer to leave it as two zones because the mathematical/coding treatment of the convective zone and diffusive zone is identical.

L154: this is often called wind pumping or just ventilation. Convection happens in winter when the surface is cold.

We have removed the references to convection.

L158: that is a very small time step! I typically run my firn air model with a timestep of one week or so. Is this needed to keep the forward Euler scheme stable?

Yes, the small timestep is necessary because of the forward Euler integration scheme and the high diffusivity of $H_2$. The model can be run at a higher timestep for other gases.

L170: Which parameterization? Goujon? Schwander? Mitchell?

We used Schwander for all three sites. We have added a footnote to Table 1 (L105).

L175: The d15N data also have thermal fractionation in them. How do you deal with this?

Thermal fractionation is not important below the upper part of the firn, and we exclude those measurements from the reconstruction due to seasonality. We have made the following revision:

L1835-187: This correction neglects the thermal fractionation of δ15N. Thermal fractionation is only important in the upper ~20 m of the firn, and these shallow measurements are excluded from the reconstructions due to seasonality (Severinghaus et al., 2001; Section 3.2).

L189: Confirm that the model is coded in volumetric concentrations, rather than in ppm. Most models work in ppm, I believe.

The model is coded in moles/m3 as noted in the text.

L197: gas phase diffusivity: do you mean free air diffusivity?

Yes, fixed.

L202: This is not really eddy mixing of course, though mathematically it is similar. This is more correctly described as dispersive mixing (Buizert and Severinghaus, 2016)

We have revised the text accordingly:

L205-207: At the top of lock-in, small, non-fractionating values of "eddy diffusivity" are prescribed to account for dispersive mixing caused by barometric pressure fluctuations (Buizert & Severinghaus, 2016).

L201-206: so the advection is coded differently in the diffusive and lock-in zones? Is it a velocity term in the former, and a box-shuffling scheme in the latter? Does this conserve mass at the boundary? Do you account for the fact that there is backflow in the lock-in zone due to compaction?

The reviewer is correct about velocity in the diffusive zone and box-shuffling in lock-in. The box shuffling prevents numerical diffusion in lock-in. To couple the two schemes, we use a "buffer" as described in the supplement of Severinghaus et al., 2010. We are impressed that the reviewer thought to ask about

backflow. We neglect the backflow term simply because turning it off yields a better empirical fit to tracers with well-constrained atmospheric histories such as CO2 and CH4. We have made the following revisions to the text:

L154-164: The UCI_2 model is largely based on Severinghaus et al. (2010). The model domain is divided into an upper "diffusive zone" and lower "lock-in zone." In the diffusive zone, vertical gas transport occurs via wind-driven mixing in the shallowest ~5 m and via molecular diffusion throughout. Diffusive mixing decreases with depth due to the increasing tortuosity of the firn. In the lock-in zone, vertical molecular diffusion ceases due to the presence of impermeable ice layers. Gas transport in the lock-in zone occurs primarily due to advection with a small non-fractionating mixing term. The model uses a forward Euler integration scheme and a time step of 324 s., There are three important differences between the Severinghaus et al. (2010) model and the UCi_2 model: 1) thermal diffusion is neglected, as it is unimportant for H2, 2) backflow due to densification in the lock-in zone is neglected in order to yield a better empirical fit to the tuning gases, and 2) our model parameterizes pore close-off fractionation differently than the Severinghaus model (see below). The model tracks the air content and composition in both open pores and closed bubbles as a function of time and depth. The model code is written and executed in MATLAB R2022a (Mathworks Inc.).

Equations 4-6: How is this implemented? The x_n and P_bubble terms occur in all three equations, so you cannot simply solve them. Is this done iteratively? Or is there a typo in the equations?

$P_{bubble}$ is a function of depth, but invariant in time and $P_{ambient}$ is constant. At the end of each time step, equations 4-6 are solved sequentially. No iteration is required.

L235: parenthesis ) missing after Rommelaere citation

Fixed

Equation 7: Can you add the arguments to the variables here to clarify? For example, G(z,t) etc

Yes, done

Eq 8: What is N? Normal distribution? What is the time step i? 1 year?

The reviewer is correct: We have made the following revisions

L259-260: Where N is the normal distribution, and $m_{atm}$ is a vector of length i that contains the discrete atmospheric H2 dry air mole fraction history with a timestep of 1 year

L267: There is no artificial smoothing, but instead the parameter beta in the autoregression. Isn't that just the same with a different name?

There difference is that we don't arbitrarily prescribe the amount of smoothing, as done in the Rommelaere method. We have made the following revision:

L272-273: 1) No arbitrarily specified smoothing criteria is imposed (instead the atmospheric history is assumed to be autocorrelated

L275: With such cut-off depths, you have only 1 to 8 m in the diffusive zones. Can you confirm?

Confirmed

L 275: Can one instead add a seasonal cycle to the atmospheric history/ inversion?

The answer to this question is a little complicated. Because of the necessity of auto-correlation (or smoothing), we would need to superimpose the seasonal cycle on the long-term trend. This would only provide additional information about the year before the firn air would sample. Those times are better constrained by atmospheric observations, so this effort would provide little scientific value.

Figures 1 and 2: is it possible to plot the firn data on the plot against their mean age?

Yes, we have added it.

L354: Can you add the Alert data to Fig. 3?

The Alert data is in Fig 3 (yellow), but we did catch a typo in the legend.

L357: Do you have a reference there for the ENSO connection? What about 1989?

We have added references for the ENSO connection and made the following addition to the text:

L392-395: The reason for the rapid decrease in observed H2 levels in the NOAA/GML data is not clear. At that time, anthropogenic emissions of H2 were likely decreasing but not rapidly enough to account for the observed decrease (Hoesly et al., 2018; Paulot et al., 2021). It is possible that the observed decrease is linked to NOAA's drifting calibration scale as discussed in Section 2.2.

L397: suggested that "the" maximum…

Fixed

L409-410: But the maximum mostly disappears when you account for the close-off fractionation

Yes, we have made the following revision:

L433-438: The late 20th century maximum in atmospheric H2 is a robust feature of these Greenland firn air reconstructions. According to the UCI_2 model, pore close-off induced enrichment is <1% at the depth of the observed H2 maximum in the lock-in zone. That is, the observed lock-in maximum is not caused by enrichment, but by a historical atmospheric maximum. However, there is evidence that the UCI_2 model underestimates the impacts of pore close-off fractionation at the Greenland sites. The implications of this underestimate for the atmospheric maximum are investigated and discussed in Section 7.

Fig. 4: is it possible to plot the firn air data against their mean (or even effective) ages on the figure? Possibly empirically corrected for close-off fractionation? I always find this extremely helpful, as it allows the reader to visualize the data density and the degree of smoothing. As an example of plotting in this style, look at the recent Ghosh et al. (2023, https://doi.org/10.1029/2022JD038281)

Yes, done

L468: Have you also compared to Mitchell et al. 2015? The Goujon and Schwander parameterizations do not account for density layering, and are commonly applied incorrectly (i.e., they were derived on cm-scale hand-samples and are applied to m-scale bulk density; because of density layering this is technically incorrect. Layering broadens the depth of bubble trapping).

See response to 3 above.

Fig. 5: It seems clear that the UCI_2 model does not build up pressure fast enough in the pores to expel fugitive gases. Shallower trapping seems like the obvious explanation to me, and tuning the closed-porosity parameterization makes sense to me as the first strategy. The Mitchell et al. 2015 parameterizations has two parameters that can be tuned.

See response to 3 above.

Fig. 5: can you also show the fit to the Antarctic dNe/N2 data somewhere? These sites are used, and so the reader will wonder whether the modeling can fit those data.

Yes, we have added it to the supplement (Figure S8) and referenced it in the text at L484.

L520+: can you give a plot of the s_c parameter? This is hard to visualize.

We have plotted s_c/s_tot in Figure 7. We prefer not to plot both because of the similarity.

Eq 13: you have different numerical advection schemes in the diffusion and lock-in zones, right? How does this impact the implementation of the bubble pressurization?

It does not. Bubble pressure is constant in each layer in time. It is calculated using quantities (new bubble volume and closed porosity) that are normalized against the volume of each layer, so the different layer thicknesses do not matter.

L547: You seek to explain the effect via the bubble compression rate only, but then alter the porosity profile after all to get something similar to the first scenario. It is the porosity tuning that makes you fit the data, not the reduced bubble compression rate – if anything the latter should make it harder to fit the data. I think this is confusing to the reader. Why not instead conclude that solely altering the pressurization rate (which this scenario ostensibly represents) does not improve the fit?

We have tried to clarify our treatment here. See response to 3 above.

Fig. 7: can you add  plots of the NEEM and Summit closed pore pressures? That seems needed to evaluate the fugitive gas enrichment.

See response to 3a

L678: what about H2 artifacts during flask storage?

We can only reasonably constrain these at NEEM due to a lack of data from other sites. For NEEM, the effects were small (~1%). In lieu of trying to constrain the blank, we increased the uncertainty on the firn air measurements by a factor of 10 (see discussion in L126-129)

---

## Author Response (AR2)

I feel obligated to ask the authors to redo the same revision again, as the revisions they make often miss the point and are incomplete. I was, and still am, quite enthusiastic about this paper and trust its results. But perhaps the authors took the positive reviews from both referees as license to perform very minimal edits to the manuscript (the editor had asked for a major revision).

Evaluating this revision has been made more challenging by the fact that the line numbers listed in the author response are incorrect (as in, the line numbers provided for the corrections do not actually match those in the text, neither the regular text nor the version with tracked changes). In other cases (such as their response to my point 3b) the authors claim to have made changes without actually doing so – the proposed "revised" text is already in the original.

All of this this does not inspire confidence in the reviewer that their comments were considered carefully. The authors will need to submit another revision of this manuscript in which they engage more deeply with the actual substance of the original review comments. Below I add clarification in response to the author's comments.

The following revisions have been made to the paper:

1) Only model results using model configurations that accurately capture neon enrichment are presented.
2) Tunu is excluded from the revised manuscript because there is no neon data from that site.
3) Instead of describing "alternative models" we instead use "model configurations" with different tuned pore close-off fractionation parameterizations.
4) We avoid physical justification for the pore close-off fractionation tunings (see below).
5) The model was updated and re-tuned to include backflow. The text and figures have been updated accordingly.

The final results (Figure 9) are essentially unchanged, but the revised methods are clearer, and the scientific reasoning is strengthened.

Regarding my original point (1)

The manuscript still feels schizophrenic. The authors agree that the "altered" scenarios are more realistic and that Figure 9 represents the best estimate of the atmospheric history of H2, yet most of the manuscript still discusses the regular UCI_2 simulation as the main result and model solution. For example, the abstract (lines 22-26), and sections 4-6 still discuss the UCI_2 scenario as if it were the main result of the paper.

As an additional note, it is confusing that the name of the firn model (UCI_2) is identical to that of the baseline scenario. I would recommend renaming the scenario. The altered scenarios are also run on the UCI_2 model – it is not the model that is at fault, but rather the calibration of the model.

I further do not agree that the altered scenarios are ad hoc. To me, in order to perform atmospheric reconstruction of H2 one needs to calibrate the firn air transport model to correctly simulate the

expulsion of fugitive gases during close-off – very much in the same way that we calibrate the diffusivity of regular gases in the diffusive zone to simulate their movement.

Only results for $H_2$ from model configurations that accurately capture neon enrichment are included in the revised manuscript. We have clarified our treatment of pore close-off fractionation in Section 3.2, and instead of referring to "alternative models," we describe different model configurations for porosity partitioning and pore close-off fractionation. The 3 model configurations discussed in Section 3.2 are:

1) Goujon et al. (2003)- this is presented as a starting point to demonstrate that "typical" partitioning between closed and open porosity will not accurately model the neon enrichment at NEEM and Summit. $s_{co}$ used for the Goujon configuration is given in Table 3
2) *Mitchell_optimized*- an empirically tuned porosity partitioning that uses the Mitchell et al. (2015) formulation. This configuration shows good agreement between modeled and measured neon at NEEM and Summit. For the remainder of the paper, we consider this the "default" configuration. But, the optimized closed porosity profile is qualitatively different from previously observed and modeled closed porosity profiles, and it is important to demonstrate that our results are not entirely dependent on a strange porosity profile.
3) *Compression*- an empirically tuned 2-stage porosity partitioning that is forced to be more "realistic" than the "*Mitchell_optimized*" configuration. In order to accurately model the neon measurements with a "realistic" closed porosity profile, we must also invoke a reduction in bubble compression (discussed further below). This configuration shows good agreement between modeled and measured neon at NEEM and Summit. We include this configuration to demonstrate that our results are insensitive to how we model pore close-off fractionation as long as it is modeled accurately.

Regarding my original point (2)

The authors did not perform the test I suggested. They show two age distributions in Fig. S7, which is unrelated to the suggestion I made. I suggested that they apply an empirical correction for close-off fractionation based on dNe/N2 and then treat the gas a as a normal non-fugitive gas. This should be very straightforward. I am curious whether it agrees with the Figure 9 reconstruction.

Pore close-off fractionation affects the age of the calculated age distribution, not just the integrated area. To properly conduct an inversion on empirically corrected $H_2$ measurements, one would need to use a normalized version of the age distribution plotted in blue in Figure S3 (i.e. one that integrates to 1 instead of 1.06). Turning off pore close-off fractionation yields the distribution plotted in pink. That is, the age distributions used to invert the empirically corrected measurements would be too young. The effect on age is somewhat more dramatic at the Antarctic sites than the Greenland sites, and we have chosen to plot Megadunes age distributions in Figure S3 to better illustrate the effect. In our earlier work on Antarctic firn air, we pursued empirically correcting the measurements before realizing the effects pore close-off fractionation has on the age. We prefer not to suggest to readers that empirical correction of $H_2$ measurements is valid. We have plotted the results of such an inversion below, together with the joint reconstructions. The Greenland results are fairly similar, but the Antarctic results are quite different.

[Figure]

Regarding my original point (3)

The updated Figure 8 (thanks for that!) sheds a lot of light on what is happening in the model. The reason these three scenarios give the same dNe/N2 solution is that they have the same bubble pressure profile R. I fear that this is very much obscured by the text which seeks to discuss the processes. In reality, the authors just tune the closed porosity until the dNe/N2 data fit. In each scenario, the authors essential increase the closed porosity at shallower depths, which increases the bubble pressure R. If I may be blunt, the discussion of physical processes is just window dressing for a fitting exercise. I am totally fine with brute force calibration/tuning, if it fits the data. Give the uncertainties in porosity this seems justified to me. If anything, the scenarios demonstrate that as long as one gets R right, one fits dNe/N2. The details of how this is done seem to be secondary.

Note: the relevant figure is now Figure 2. We have clarified the treatment in section 3.2. We have largely jettisoned the discussion of physical processes in favor of tuning the porosity partitioning. In the *Compression* model configuration, we still optimize the rate of bubble pressurization, but that is just a convenient way to model the neon data. The revised manuscript describes the rate of bubble pressurization as a tuning lever. We direct the reviewer towards the following passages:

L276-277: Here, we use the rate of bubble pressurization as an additional tuning lever to examine the sensitivity of the model results to the specific physical parameterizations in the model.

L295-296: The alternative model configurations are considered empirical tuning methods to fit the Ne data and should not be used to draw conclusions about the underlying firn physics.

L662-663: The *Mitchell_optimized* and *Compression* model configurations are empirical tuning methods and do not address uncertainty around the physics underlying the observed pore close-off induced enrichment in Greenland firn air

In my comment 3b I was really suggesting that the "reduced compression" scenario is basically mislabeled, as the real reason the fit is improved is the closed porosity parameterizetion (the reduced compression actually makes things worse). By changing two parameters at once, the test is hard to interpret. In their response they claim to clarify the text, but I see no changes to section 7.2. Such textual clarifications (which were not made, as far as I can tell) also do not fix the bigger issue of mislabeling and/or unclear experimental design (maybe I should have been more explicit about this). The reader will come away from this section thinking that reduced compression somehow improves the simulation of fugitive gases, but in fact the opposite is true. Fig. 8 shows that to compensate for the reduced compression, the authors have to increase the shallow trapping even more such that the R is correct. There MAY be reduced compression of bubbles, but the tests performed cannot say anything about this as they compensate with changes to the porosity profile.

The revised manuscript clarifies this point. The reduced bubble compression (now the *Compression* configuration) is simply a way to model the data using a fairly realistic closed porosity profile. The problem, broadly described, is that with a realistic closed porosity profile, R increases too rapidly with depth to capture the neon data. That is why we cannot use only adjust $s_{co}$ for the Goujon parameterization or $\rho_{co}$ for the Mitchell parameterization. To match the neon data, either bubble close-off must take place over a broader depth range (as in the Mitchell_optimized configuration), or we must invoke some other process. Here, we have chosen the rate of bubble compression as a convenient lever (as mentioned above). We first describe our attempt to model neon with a realistic porosity partitioning. Only when that fails do we introduce reduced bubble compression:

L265-278: "We also examined one additional model configuration, referred to as the "Compression" configuration, in which we force the closed porosity profile to have a more typical complete close-off depth. In this configuration, the closed porosity above some optimized critical depth (zcrit) is parameterized as in Goujon et al. (2003; their equation 9) with an optimized sco parameter. Below the critical depth, closed porosity increases linearly with depth, reaching complete close-off at the same depth as in the Goujon et al. (2003) base-case (79.2 m at NEEM and 81.5 m at Summit). A similar two-stage closed porosity parameterization was used in Severinghaus & Battle (2006). The resulting closed porosity profiles are more similar to previously published profiles and probably more realistic (Figure 2). However, even with an optimized zcrit and sco, we found that R increases too rapidly with depth to capture the δ22Ne/N2 at both the top and the bottom of the lock-in zone. Therefore, to generate the necessary R profile while maintaining a realistic closed porosity profile, some other physical process must be modified in the model in addition to the closed porosity profile. Possible candidates include the rate of pressurization of bubbles, the rate of densification, and the rate of bubble close-off. A detailed observation-based investigation of these processes at these sites would require field data that do not currently exist. Here, we use the rate of bubble pressurization as an additional tuning lever to examine the sensitivity of the model results to the specific physical parameterizations in the model."

It is important to include this configuration to demonstrate our results are not dependent on the strange *Mitchell_optimized* closed porosity profile. We agree that the tuned model configurations cannot say anything about the underlying firn physics and direct the reviewer to the following passages:

L295-296: The alternative model configurations are considered empirical tuning methods to fit the Ne data and should not be used to draw conclusions about the underlying firn physics.

L662-663: The *Mitchell_optimized* and *Compression* model configurations are empirical tuning methods and do not address uncertainty around the physics underlying the observed pore close-off induced enrichment in Greenland firn air

In comment 3c I was suggesting that the NEEM closed porosity parameterization they start out with (the UCI_2 run) is not reliable. They respond by adding a comment that the Mitchell parameterization didn't improve the fit - not exactly a full response. What does "not optimal" mean here? Surely it performed better than the UCI_2, and perhaps by tuning the 3 parameters in this parameterization one can also get a good fit to the dNe/N2 data?

We are now using an optimized Mitchell parameterization as the default model configuration. The optimized parameters are very different from what Mitchell recommends, and, as shown in Figures 2a and 2c, the optimized closed porosity profile is qualitatively different from "typical" closed porosity profiles with bubble close-off taking place over a much broader depth range and a deeper complete close-off depth.

L259-264:  The *Mitchell_optimized* parameters are significantly different from the recommendations of Mitchell et al. (2015). Furthermore, the resulting closed porosity profiles are qualitatively different from other measured and modeled closed porosity profiles and are probably not physically realistic (Figure 2). Bubble close-off takes place over a much broader depth range in the *Mitchell_optimized* configuration, and complete close-off (i.e. $s_c > 0.999 s_{total}$) does not occur until depths of 111.0 m (NEEM) and 112.5 m (Summit). Firn air could only be sampled to a depth of 75.9 m (NEEM) and 80.1 m (Summit), suggesting that complete close-off is actually significantly shallower than in the Mitchell_optimized configuration.

In my comment 3d, please add some text to the discussion stating that as long as one gets R right, one gets the dNe/N2 right.

L245-252:  At equilibrium, modeled enrichment of Ne and $H_2$ is controlled primarily by the ratio of the volume-weighted average pressure in the open and closed pores to the ambient pressure, adjusted for mixing. We define a new parameter ($R$) to describe this ratio:

$$R = (P_{bubble} s_c / s_{total} + P_{ambient} s_o / s_{total}) / P_{ambient} \tag{8}$$

When previously published parameterizations of partitioning between open and closed pores are implemented in the model, $R$ begins to increase too deep in the firn to capture the shallower $\delta^{22}Ne/N_2$ measurements (Figure 1 and Figure 2).

In response to my comment 3e, could you please just extend the plot further down? The alternative scenarios provide the best atmospheric reconstruction (by the author's admission), so it seems that we

should have the details to evaluate them. None of them go all the way to a closed pore fraction of 1, which is what prompted my curiosity

For the Goujon et al. (2003) base-case and the *Compression* configuration, Figure 2 now shows a closed porosity ratio of 1. The *Mitchell_optimized* configuration reaches 1 much deeper, and we would prefer not to plot it here as it is probably not realistic. In the text we have listed the complete close-off depth for all three configurations.

L260-264: Furthermore, the resulting closed porosity profiles are qualitatively different from other measured and modeled closed porosity profiles and are probably not physically realistic (Figure 2). Bubble close-off takes place over a much broader depth range in the *Mitchell_optimized* configuration, and complete close-off (i.e. sc>0.999stotal) does not occur until depths of 111.0 m (NEEM) and 112.5 m (Summit).

L268-269: Below the critical depth, closed porosity increases linearly with depth, reaching complete close-off at the same depth as in the Goujon et al. (2003) base-case (79.2 m at NEEM and 81.5 m at Summit)

The response to my Eq. 4-6 comment is insufficient. The way the equations are written they cannot be solved sequentially. So this needs to be clarified.

We have added additional subscripts to equations 5 and 6 and clarified the text.

(L211-219):

$$x_{n(bubble)\_eq} = P_n/P_{bubble} \hspace{5cm} (5)$$

$$x_{n(firn)\_eq} = P_n/P_{ambient} \hspace{5cm} (6)$$

Where $P_n$ (Pa) is partial pressure of gas $n$ ($H_2$ or Ne; See section 5), $x_n$ is mole fraction of gas $n$, $P_{bubble}$ (Pa) is the total bubble pressure, $P_{ambient}$ (Pa) is the ambient pressure in the open pores, and $s_c$ is closed porosity. The subscripts *bubble* and *firn* distinguish between the closed and open porosity. Equation 4 is executed at the end of each time step in each grid cell in the model. Then the mole fraction of gas $n$ in the bubbles and firn air is updated to its equilibrium value using equations 5 and 6 before proceeding to the next timestep ($x_{n(bubble)\_eq}, x_{n(firn)\_eq}$).

In their response to L201-206, they neglect a critical process (backflow) because it improves the fit to two tracers - however, the correct approach here would be to include the physical process and calibrate the diffusivity profile to fit the tracers. They claim the model tracks air content, but this cannot be correct if you neglect the backflow.

We updated the model using equation A27 in Severinghaus & Battle (2006) and re-tuned. The text and figures have been updated accordingly.

In their response on Line 267. How do they pick the beta parameter? My main point is that instead of an arbitrary smoothness parameter, they impose a autocorrelation parameter – can you explain why this is less arbitrary? Isn't the beta parameter effectively a smoothness parameter also?

Beta is not explicitly prescribed. The source of concern may have been this sentence: " $\beta$ is a positive scalar which may be specified or varied as a free parameter." In the results presented here, Beta is only varied as a free parameter. Therefore, we have deleted the sentence. Additionally, we have changed $\beta$ to $\alpha$ so that our nomenclature directly aligns with Aydin et al., (2020).

The beta parameter quantifies how much atmospheric $H_2$ can change from one year to the next and can take on a range of values that allow an ensemble of different trajectories for $H_2$ with different variances in time. Larger values of beta result in higher the year-to-year change in $H_2$ in each atmospheric trajectory, with no direct relevance to the smoothness of the posterior distributions, i.e. the presented firn inversion results. The smoothness directly emerges out of the structure in the measurements and the magnitude of the associated error bars, and for multi-site inversions, also the agreement between the data from different sites. The MCMC optimization algorithm selects the most likely Beta from a uniform prior given the firn air data. In this way, Beta depends on the firn air data instead of an arbitrary user-selection.